**TOPICAL REVIEW**

# The mysterious middlemen making your vision pop: understanding the function of amacrine cells

Victor Calbiague-Garcia, Deborah Varro, Thomas Buffet and Olivier Marre

*Institut de la Vision, Sorbonne University, CNRS, INSERM, Paris, France*

Handling Editors: Laura Bennet & Karin Dedek

The peer review history is available in the Supporting Information section of this article (https://doi.org/10.1113/JP287958#support-information-section).

**Abstract figure legend** This review aims to illustrate the diversity and function of amacrine cells in the retina. The diversity of amacrine cell subtypes is depicted based on morphology, stratification and neurotransmitter expression, along with their synaptic connectivity with bipolar and ganglion cells, emphasizing inhibitory and modulatory roles. Advances in transcriptomic and genetic tools have deepened our understanding of amacrine cell diversity, uncovering a wide range of interactions, such as feedback and feedforward inhibition (or excitation) and disinhibition. Technological advances have also revealed their role in key retinal circuits, e.g. direction selectivity, contrast adaptation, approaching motion, object motion and anticipation.

V. Calbiague-Garcia and D. Varro, contributed equally to this work.

**Abstract**　In many brain regions, inhibitory interneurons represent a highly diverse class of cells, and the specific roles of most subtypes remain unclear. This diversity is particularly striking in the retina, where amacrine cells, the primary inhibitory interneurons, form the most diverse population, with nearly 67 subtypes in mice. Recent methodological advances have provided unprecedented insight into this complexity. Techniques such as transcriptomics, connectomics and targeted electrophysiological recordings have made it possible to isolate and characterize individual amacrine cell types. Here, we review current knowledge of amacrine cells and discuss how emerging approaches are advancing our understanding of their function, with a focus on the mouse retina. Several subtypes can now be genetically targeted, allowing for detailed study of their morphology and light responses. A promising avenue of research is investigating how these cells process complex stimuli and whether their responses vary across different dendritic compartments. Amacrine cells play a fundamental role in visual computations, often through dedicated circuit motifs. However, for most subtypes, their specific contributions to these motifs remain unknown. A key open question is whether different amacrine subtypes function as independent units within distinct circuits or if they are interconnected within a broader, recurrent inhibitory network. Answering this will be essential to understand how amacrine cells contribute to retinal processing fully.

(Received 29 December 2024; accepted after revision 31 March 2025; first published online 18 April 2025)

**Corresponding author** V. Calbiague-Garcia: Institut de la Vision, Sorbonne University, CNRS, INSERM, Paris, France. Email: victor-manuel.calbiague-garcia@inserm.fr

## Introduction

The retina is a specialized neural tissue in which visual signals are transduced by photoreceptors into electrical impulses and then processed by various retinal cells. Rather than merely functioning as a passive camera that captures light for the brain to interpret, the retina performs intricate computations through its multiple cellular layers (Gollisch & Meister, 2010).

A key feature of the retina is its neuronal diversity. In the mouse retina alone, approximately 120 distinct cell types have been identified (Fig. 1*B*), including photoreceptors, horizontal cells, bipolar cells, amacrine cells and ganglion cells (Demb & Singer, 2015; Euler et al., 2014; Masland, 2012; Shekhar et al., 2016). These cells are organized into layers (Fig. 1*A*), forming a pathway in which visual information flows from photoreceptors to bipolar cells and ultimately to ganglion cells, which transmit spike train outputs to the brain (Kerschensteiner, 2022). At the same time, horizontal cells and amacrine cells (ACs), two types of interneurons found in two retinal plexiform layers, influence the final output by modulating communication between bipolar and ganglion cells (GCs) through presynaptic and postsynaptic inhibition

ACs are a diverse group of inhibitory interneurons that shape the spatial and temporal properties of visual signals through feedforward, feedback, lateral and crossover inhibition. They play a crucial role in encoding complex features of the visual environment (Fig. 2) (Diamond, 2017; Gollisch & Meister, 2010). The importance of ACs in visual signalling is reflected by their diversity – mice possess approximately 60 AC subtypes (Fig. 1) (Helmstaedter et al., 2013; Li et al., 2024; Peng et al., 2019; Yan et al., 2020) – making them the most diverse neuronal population within the retina. Paradoxically, despite their essential function in retinal computations, the roles and connectivity of most AC subtypes remain poorly understood, with only a handful of subtypes being characterized

**Víctor Calbiague-García** is a biologist who earned his PhD in neuroscience under the supervision of Oliver Schmachtenberg at the Interdisciplinary Center for Neuroscience, University of Valparaíso, Chile. His research focused on understanding how the physiology and metabolism of the inner retina changes under normal and pathological conditions. He is currently a postdoctoral researcher with Olivier Marre at the Vision Institute in Paris, where he investigates how the different subtypes of amacrine cells shape ganglion cell responses. **Deborah Varro** completed her PhD under the supervision of Olivier Marre at the Vision Institute in Paris in late 2024. Since then, she has continued her research as a postdoctoral fellow in the same laboratory. Both researchers share a common interest in understanding how the retina processes visual information, with a particular focus on the physiology and function of the inner retina.

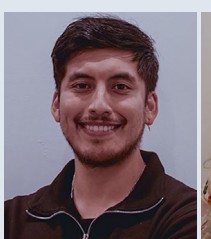
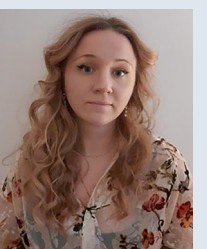

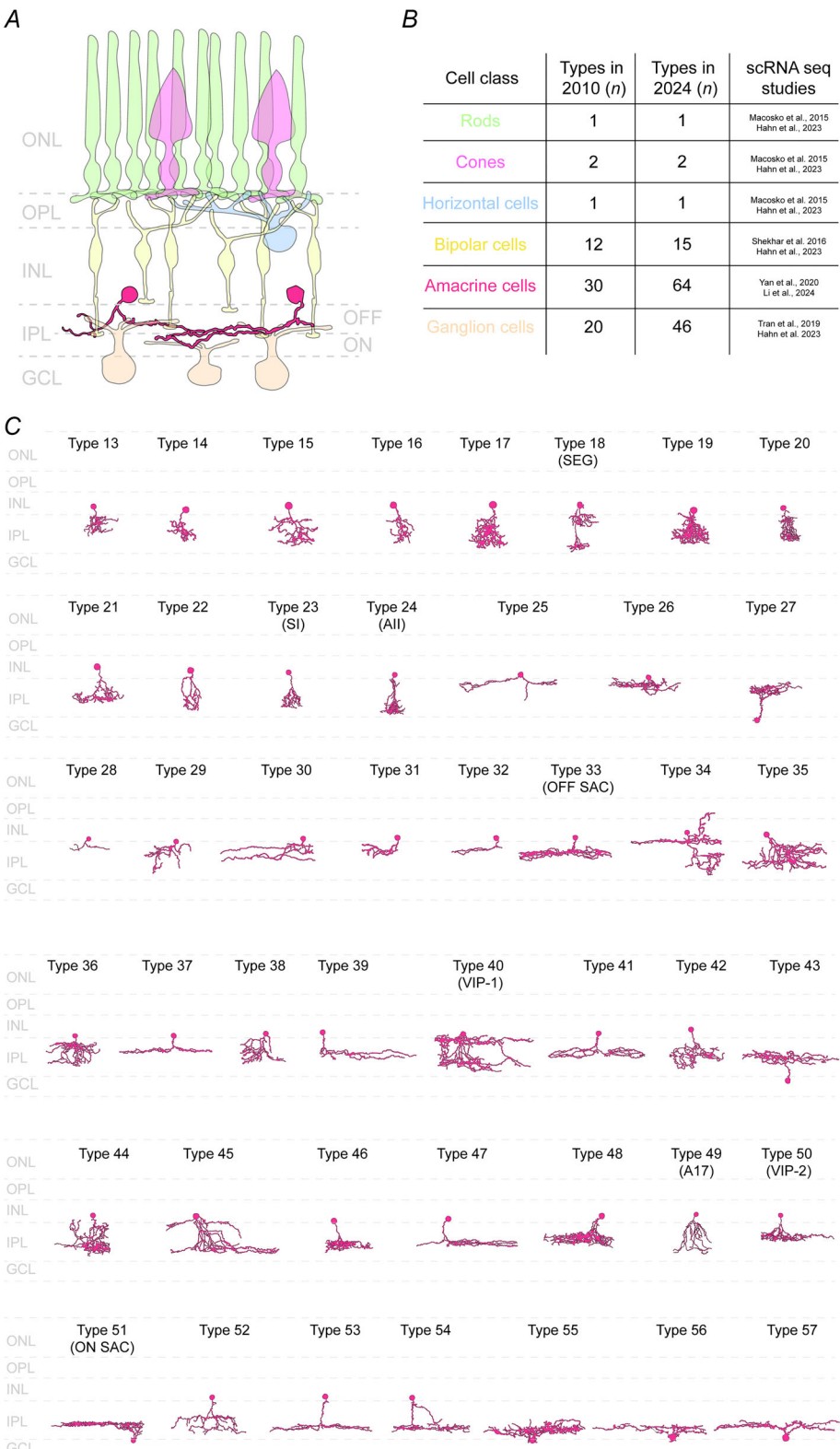

**Figure 1. Diversity of amacrine cells in the mouse retina**

*A*, organization of the retina, showing the position of the amacrine cells. The somas of the amacrine cells are positioned in the inner plexiform layer (INL), and project to the inner plexiform layer (IPL) where they make synapses with bipolar cells, ganglion cells and other amacrine cells. *B*, table summarizing how the diversity of different retinal cell classes has changed over the past 14 years based on transcriptomic studies. Among all retinal classes,

amacrine cells show the greatest diversity in molecular terms. *C*, morphological diversity of the amacrine cells based on Helmstaedter et al. (2013). Some types also show the specific amacrine cell to which they have been related. GCL, ganglion cell layer; INL, inner nuclear layer; IPL, inner plexiform layer; ONL, outer nuclear layer; OPL, outer plexiform layer.

(Euler et al., 2002; Grimes et al., 2010; Jacoby et al., 2018; Kim et al., 2015; Lee et al., 2016; Park et al., 2015, 2018; Vaney et al., 2012).

How can we study and understand the role of such a diverse population of cells? This requires integrating structural, molecular, genetic and functional information. A cell type is typically defined by its distinct computational role, morphology and expression of specific proteins. ACs, like other retinal cell classes, have unique characteristics that shape their functions within the visual circuit. For example, various shapes of ACs correspond to unique protein combinations that influence their physiological properties and response patterns. Each AC type exhibits a particular gene expression pattern, a morphology with defined neurite stratification, specific synaptic transmitters, and selective synaptic partners within the retinal network. All of that defines the connectivity of a type of AC, which, in turn, determines its role in retinal visual processing, such as motion detection or contrast sensitivity.

This review provides a comprehensive overview of AC diversity and its crucial role in visual processing. We begin by exploring the structural and molecular features that distinguish AC subtypes, emphasizing how genetic and transcriptomic advancements have revolutionized our ability to classify and study these cells. Building on this, we examine the functional contributions of ACs in the retina, exploring the different circuits they form and their impact on specific visual computations.

## How diverse are the amacrine cells?

ACs in the retina exhibit the most remarkable variety of shapes and structures (Fig. 1*C*). Their distinct anatomy reflects their specialized synaptic connections, enabling them to participate in distinct visual processing tasks. In this section, we describe the three morphological characteristics of ACs: their dendritic tree, their stratification and their spatial distribution. Together, these three features determine each AC type's specific partner and function.

**Diversity in morphology of amacrine cells.** The importance of cellular morphology in the retina was first emphasized by Santiago Ramón y Cajal, who laid the groundwork by distinguishing the main retinal cell types based on their anatomy (Cajal, 1893). Since then, studies such as those by MacNeil and Masland (1998) have expanded our knowledge and identified several AC subtypes based purely on their shapes and dendritic patterns. Before the development of genetic tools, AC classification relied primarily on dendritic morphology. However, it is essential to mention that early immunohistochemical techniques further improve this classification by using molecular markers – such as cholinergic, serotonergic, vasoactive intestinal peptide (VIP), nitric oxide synthase (NOS), substance P, and dopamine – to distinguish AC subtypes beyond their anatomical features.

Currently, between 40 and 50 AC types have been identified through morphological studies (Badea & Nathans, 2004; Helmstaedter et al., 2013). These ACs exhibit considerable diversity, from the positioning of their cell bodies within retinal layers to the breadth and complexity of their dendritic tree (Fig. 1*C*)

While most ACs reside in the innermost part of the inner nuclear layer (INL), some exceptions exist. A subset known as 'displaced ACs' have cell bodies located within the inner plexiform layer (IPL) and even the ganglion cell layer (GCL) (Helmstaedter et al., 2013). Based on morphology, around 10 displaced AC types have been identified (Binggeli & Paule, 1969; Pérez de Sevilla Müller et al., 2007; Perry & Walker, 1980). However, recent advances in spatial transcriptomics have expanded this number to at least 12 (Choi et al., 2023). Although certain displaced ACs, such as ON starburst ACs (Fig. 1*B*, type 51), have been recognized through morphological studies, most cannot be classified based on morphology alone. Instead, many AC subtypes require a combination of morphological and genetic tools for accurate identification. This integrative approach has facilitated the identification of additional displaced ACs, including the corticotrophin-releasing hormone (CRH) ACs, neuronal nitric oxide synthase type 2 (nNOS2) ACs, SST ACs, VIP-3 ACs, a subset of B/K ACs and S3-Gbx2 ACs (Table 1).

*Dendritic tree size of amacrine cells.* ACs process visual information primarily through their dendritic trees, which vary widely in size. Based on their dendritic arbour diameter, ACs are classified in three categories: narrow-field ACs (dendritic arbour diameter <125 μm), medium-field ACs (125–400 μm) and wide-field ACs (>400 μm) (Badea & Nathans, 2004; Lin & Masland, 2006; MacNeil & Masland, 1998; Pérez de Sevilla Müller et al., 2007). Based on their arbours, 12 types of ACs have been identified as narrow-field and 33 types as medium/wide-field ACs (Helmstaedter et al., 2013) (Fig. 1*C*). Narrow-field ACs have compact dendritic arbours, allowing for localized information processing,

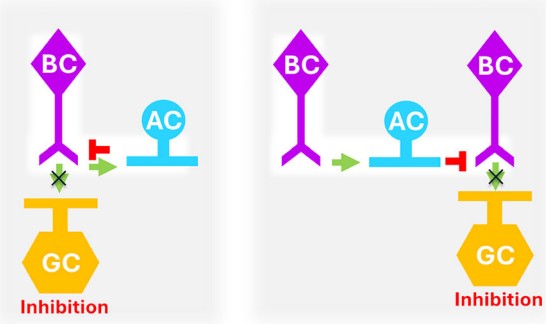

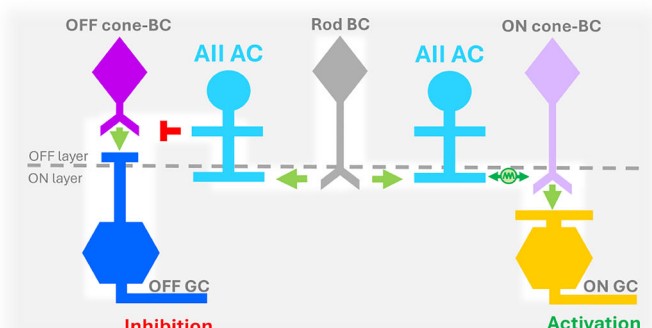

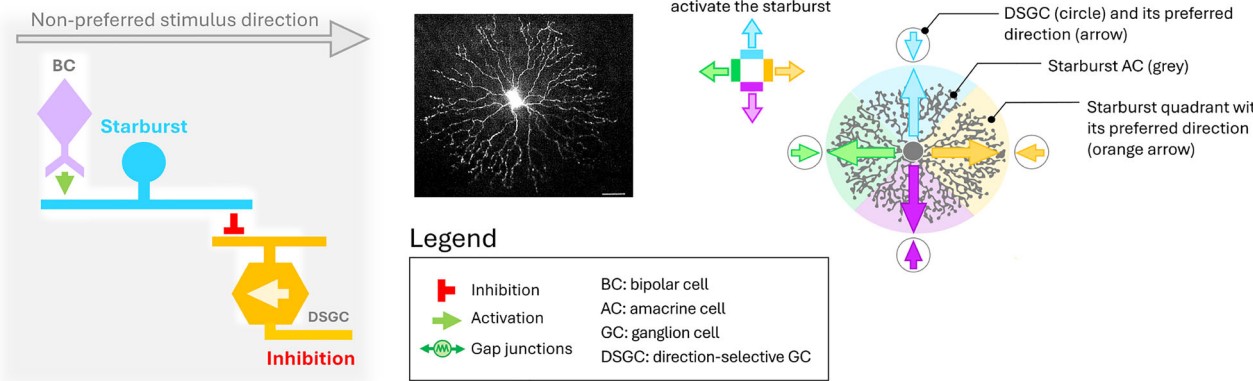

**Figure 2. Schematic representation of vertebrate retinal circuitry**

Photoreceptors and horizontal cells have been omitted most of the time for clarity. *A*, surround inhibition occurs when ACs located far away from the GC are activated by the retinal circuit above that responds to a widespread visual stimulus. *B*, an AC disinhibits the target of another AC (here a GC but can also be BCs), allowing it to be depolarized in response to a visual stimulus. *C*, feedforward inhibition from an AC activated by BC to a GC. Horizontal cells can also form feedforward inhibitory synapses with bipolar cells (not shown). *D*, an AC inhibits the very BC that excites it (reciprocal feedback inhibition) or (shown here) inhibits another BC of the same type as the one that excites it (non-reciprocal feedback inhibition). *E*, crossover inhibition from an AC (the AII AC) activated by a BC. Interaction between rod and cone (ON and OFF) pathways, leading to the inhibition of OFF GCs and activation of ON GCs. *F*, schematic representation of the mechanisms governing the organization of direction-selective circuits in the retina. Left: schematic representation of the inhibitory connection between starburst AC and DSGC. DSGCs typically generate strong action potentials in response to motion in their preferred direction. In the schema, the orange arrow indicates the preferred direction of the DSGC. However, when motion occurs in the opposite direction of their preferred one (grey arrow), DSGCs are inhibited by Starburst ACs. Those starbursts receive

excitatory glutamatergic inputs from BCs and send inhibitory GABAergic inputs to DSGCs. Middle: flatmount view of a luciferin-injected starburst AC in the central retina showing a round soma and symmetrical dendritic branching organization. The distal ends of the dendrites bear numerous small swellings, which are rarely seen in the proximal dendrites. Taken from Tauchi and Masland (1984). Right: top view of starburst AC showing its asymmetrical inhibition of four different direction-selective GCs (DSGCs). Each quadrant of the starburst preferentially forms GABAergic synapses with the DSGC, whose preferred direction is opposite the direction that activates that particular quadrant of the starburst. Adapted from Chen and Wei (2018).

whereas wide-field ACs integrate visual signals across large retinal areas, playing a key role in global processes such as surround suppression (Masland, 2012).

Unlike most neurons, the majority of ACs lack a typical long axon, a characteristic reflected in their name – 'amacrine,' meaning 'without a long fibre' in Greek. However, there are exceptions. Völgyi et al. (2001) identified six types of 'polyaxonal ACs', and Lin and Masland (2006) later found that about two-thirds of the wide-field ACs they studied exhibited axon-like processes. These diverse morphologies allow ACs to integrate and relay visual information across different targets and over a broad visual field.

*Stratification of amacrine cells.* The retina is organized into distinct layers (Fig. 1*A*), where the positioning of a cell's processes often reflects its synaptic connections. In the IPL, each sublayer hosts specific sets of pre- and postsynaptic partners, meaning AC morphology is closely linked to distinct synaptic interactions and, consequently, specific functional roles.

The IPL is divided into five sublayers corresponding to the ON and OFF visual pathways. These layers allow ACs to specialize in processing either light increments (ON) or light decrements (OFF) (Fig. 1*A*). The stratification of starburst ACs (Types 33 and 51, Fig. 1*C*) often serves as a reference for defining the boundaries of these sublayers, since these ACs are found in both OFF (sublayer 2, closer to INL) and ON (sublayer 4, closer to GCL) regions. These sublayers are labelled using choline acetyltransferase (ChAT), a molecular marker for starburst ACs.

ACs exhibit diverse stratification patterns within the IPL, classifying them as monostratified, bistratified, or pluristratified. Monostratified ACs extend their processes within a single IPL sublayer, forming connections primarily with neurons confined to that layer. In contrast, bistratified ACs project across two distinct sublayers, allowing them to integrate signals from both the ON and OFF pathways. This ability to bridge retinal circuits makes bistratified ACs important for complex visual computations, such as contrast detection, motion sensing and crossover inhibition (Masland & Raviola, 2000). Pluristratified ACs, on the other hand, extend their processes across multiple IPL sublayers, further increasing their capacity for signal integration. Narrow-field ACs frequently exhibit this pattern – for example, AII ACs (Fig. 1*C*, Type 24) (Marc et al., 2014) – whereas wide-field ACs are more commonly monostratified, such as the

tyrosine hydroxylase type 2 (TH2) ACs (Brüggen et al., 2015). However, exceptions exist. Some polyaxonal ACs, for instance, project across several IPL layers, forming specialized circuits (Jia et al., 2020; Pérez de Sevilla Müller et al., 2007). While ACs typically form synapses at the ends of their dendrites, some also establish connections along the length of their processes, known as *en passant* synapses. These unconventional synapses add another layer of complexity to retinal circuitry, allowing for additional interaction points beyond the traditional synaptic terminals. For example, ON bipolar cells have been shown to form *en passant* synapses with M1 ganglion cells and dopaminergic ACs (Dumitrescu et al., 2009; Kim et al., 2012), suggesting that AC connectivity is more complex than previously thought. These diverse morphological patterns underline the functional versatility of ACs in retinal signal integration.

**Synaptic diversity of amacrine cells.** ACs communicate with GCs and other retinal cells through a variety of synapses, using both inhibitory and excitatory neurotransmitters, as well as neuromodulators.

*Inhibitory synapses.* Most ACs primarily use GABA and glycine as inhibitory neurotransmitters (Korympidou et al., 2024; Matsumoto et al., 2024; Yan et al., 2020). The retina contains various GABA receptors (GABA-A, -B and -C), along with glycine receptors, each contributing to distinct inhibitory interactions (Gustincich et al., 1999; Koulen et al., 1996). In general, wide-field ACs are predominantly GABAergic, while narrow-field ACs mainly release glycine, although also some GABAergic narrow-field ACs can be found (Chen et al., 2011; Menger et al., 1998; Pourcho & Goebel, 1985).

*Excitatory synapses.* While ACs are primarily inhibitory, some release excitatory neurotransmitters like glutamate and acetylcholine. For example, glutamatergic ACs (vesicular glutamate transporter 3 (VGluT3) ACs) release glutamate to create excitatory circuits in the retina (Lee et al., 2014, 2016), while starburst ACs use acetylcholine to communicate with direction-selective GCs (Masland & Livingstone, 1976).

*Neuromodulatory synapses.* In addition to neurotransmitters, ACs release neuromodulators such as dopamine, nitric oxide, endocannabinoids, VIP, etc. (Popova, 2014; Vaney, 2004; Yazulla, 2008) (see 'Amacrine

**Table 1. ACs identified in the literature and different physiological properties**

| AC type | Subtypes | AC cluster (from Yan et al. 2020) | AC cluster (from Matsumoto et al. 2024) | Specific genes (protein coding) | Mouse line | Canonical neuro-transmitter | Co-release | Polarity | Dendritic field | Stratification | Soma localization |
|---|---|---|---|---|---|---|---|---|---|---|---|
| AII (Yan et al., 2020; Demb & Singer, 2012; Marc et al., 2014) | — | C3 | — | *Gjd2, ProX1, Dab1, Nfia, Dner* | — | Glycine | — | ON | Narrow | Multistratified | INL |
| Starburst (Helmstaedter et al., 2013; Yan et al., 2020; Pottackal et al., 2020; Wei, 2018; Famiglietti, 1991) | ON and OFF | C17 | G10 (ON) and G37 (OFF) | *Chat* | *ChAT-Cre* | GABA | Ach | ON or OFF | Medium | Monostratified | INL, GCL |
| A17 (Yan et al., 2020; Nelson & Kolb, 1985; Grimes et al., 2010; Grimes et al., 2010) | — | C6 | G18 | *Prkca (PKCα), Sdk1, Calb2, Dab1* | — | GABA | — | ON | Wide | Multistratified | INL |
| VGluT3 (Yan et al., 2020; Lee et al., 2016; Tien et al., 2016; Grimes et al., 2011; Haverkamp & Wässle, 2004) | — | C13 | — | *Slc17a8 (VGlut3)* | *Slc17a8-IRES2-Cre* | Glycine | Glutamate | ON-OFF | Narrow | Bistratified | INL |
| TH2 (Yan et al., 2020; Brüggen et al., 2015; Kim & Kerschensteiner, 2017) | ? | ? | G22 | *Th2* | *TH::GFP* | GABA | Dopamine | ON-OFF | Wide | Multistratified | INL |
| CRH (Zhu et al., 2014; Park et al., 2018) | CRH-1 CRH-2 CRH-3 | C37 | G14 G31 ? | *Crh* | *CRH-ires-Cre* | GABA | — | ON | Medium Medium/wide Wide | Monostratified Monostratified | GCL GCL GCL |
| nNOS (Zhu et al., 2014; Yan et al., 2020) | nNOS1 nNOS2 | C48 and C54 | G31 G30 | *Adra1b, Tmem163* | *nNOS-CreER* | GABA | — Nitric oxide | ON ON-OFF | Medium Wide | Multistratified Multistratified | INL GCL |
| VIP (Zhu et al., 2014; Akrouh & Kerschensteiner, 2015; Yan et al., 2020; Casini & Brecha, 1991) | VIP-1 VIP-2 VIP-3 | C47, C22 and C26 | G32 ? ? | *Slc38a5, Cbln4, Abi3bp* | *VIP-ires-Cre* | GABA | VIP | ON-OFF ON ON | Wide Narrow Medium-wide | Bistratified Monostratified Monostratified | INL INL GCL |
| SST (Zhu et al., 2014) | SST-1 ? | C63 | G3 | *Sst, Gal* | *Penk-Cre SST-ires-Cre* | GABA | — | ? | Wide | Multistratified | GCL |

*(Continued)*

**Table 1. (Continued)**

| AC type | Subtypes | AC cluster (from Yan et al. 2020) | AC cluster (from Matsumoto et al. 2024) | Specific genes (protein coding) | Mouse line | Canonical neurotransmitter | Co-release | Polarity | Dendritic field | Stratification | Soma localization |
|---|---|---|---|---|---|---|---|---|---|---|---|
| B/K (Park et al., 2024) | ? | ? | ? | Bhlhe22 | Bhlhe22-Flpo x KOR-Cre | GABA | — | ON and OFF | Wide | Multistratified | INL and GCL |
| nGnG (Yan et al., 2020) | nGnG-1 | C24 | — | Neurod6, Ebf3, Ppp1r17 | Neurod6-Cre | ? | ? | ON-OFF | Narrow | Multistratified | INL |
| | nGnG-2 | C10 | | Cntn6, Cck, Ebf3, Nfix | Cntn6-lacZ-Cre Cck-IRES-Cre Cntn5-Cre | | | | | | |
| | nGnG-3 nGnG-4 | C30 C36 | | Cntn5, Ppp1r17 Gbx2, Lhx9 | Cntn5-LacZ Gbx2-Creer-GFP | | | | | | |
| SEG (Yan et al., 2020; Kay et al., 2011) | ? | C4 | — | Satb2, Ebf3, Neurod6, Prom1, Nfix, Igf1 | Neurod6-cre | Glycine | — | ON | Narrow | Multistratified | INL |
| GHRH (Yan et al., 2020) | ? | C51 | — | ? | ? | ? | ? | ? | ? | ? | ? |
| PENK (Yan et al., 2020) | ? | C35 and C59 | ? | Ppp1r17, Car3l | Penk-IRES2-Cre | GABA and Glycine | ? | ? | Narrow/ medium | Multistratified | INL |
| SRIF (Vuong et al., 2015) | ? | ? | ? | ? | ? | ? | Somatostatin | ? | Wide | Monostratified | INL |
| S3-Gbx2+ (Kerstein et al., 2020) | ? | ? | ? | Gbx2+ | Gbx2$^{CreERT2-IRES-EGFP}$ | — | — | ON-OFF | Medium | Monostratified | GCL |
| SI-AC (Jo, Deniz, Xu et al., 2023) | ? | ? | — | secretogranin II | VGAT-iCreER; Scg2-tTA;Ai93 | Glycine | — | OFF to light flashes | Narrow | Multistratified | INL |
| COMS-AC (Jo, Deniz, Cherian et al., 2023) | ? | ? | — | Camk2a | VGAT-Cre(iCreER)/ Camk2a-tTA | Glycine | — | OFF | Narrow | Multistratified | INL |
| PAS4/5 (Jia et al., 2020) | PAS4/5-1 PAS4/5-2 | ? ? | ? ? | ? ? | ? ? | | | OFF ON | Wide Wide | Monostritified Monostratified | GCL GCL |

The table also includes the correspondence between them and the ACs clusters in Yan et al. (2020) and Matsumoto et al. (2024). The references used to create this table are listed under each AC name. A question mark (?) is added if no information was found.

cell synapses'). These substances act more slowly and influence a broader area, enabling more complex modulation of visual signals. For example, dopaminergic ACs regulate the strength of gap junctions, adjusting retinal circuits to different lighting conditions (Popova, 2014). Additionally, many ACs can release more than one type of neurotransmitter, allowing them to play multiple roles within retinal circuits.

**Diversity of amacrine cells in the expression of specific proteins.** As we have already pointed out, ACs have long posed a significant challenge for study due to their remarkable heterogeneity in morphology, function and interactions with other retinal cell types. Traditional approaches, such as histological and electrophysiological techniques, provided valuable insights but were often insufficient for fully characterizing the full diversity of AC types or selectively targeting specific populations for functional studies. Recent advances in genetic tools, including transcriptomics and single-cell RNA sequencing (scRNA-seq), have revolutionized the field, enabling high-resolution molecular profiling of ACs and uncovering distinct subtypes and their gene expression signatures. These breakthroughs, along with the development of targeted mouse models, now allow the identification and manipulation of specific AC subtypes with unprecedented precision.

This section focuses on two key areas: the molecular identification of retinal ACs using transcriptomic approaches and the genetic mouse models and promoters available for targeting and studying these cells. These advances offer promising avenues to further our understanding of retinal circuitry and AC functional roles.

*Molecular diversity of amacrine cells.* Recent molecular profiling studies, especially using transcriptomic approaches like bulk RNA sequencing and, more recently, scRNA-seq, have revealed that specific gene expression patterns can be used to further classify ACs into distinct subtypes (Yan et al., 2020), which is essential for studying a population as diverse as they are.

Each AC subtype expresses unique gene patterns, transcription factors, synaptic proteins, ion channels and neurotransmitters, reflecting their specific functions in the retinal network. Those molecular profiles are correlated with morphological features, such as dendritic tree structure, stratification within the IPL and functional roles within retinal circuits (Marc et al., 2013; Peng et al., 2019; Yan et al., 2020). For example, the expression of some synaptic proteins provides a detailed view of synaptic connectivity: synaptic proteins like cytomatrix proteins identify presynaptic terminals and their distribution within the retina (Brandstätter, 1999), while PSD-95 protein provides information about the postsynaptic architecture of ACs (Koulen et al., 1998). Additionally,

calcium channel expression – which mediates calcium influx, crucial for neurotransmitter release and synaptic modulation – helps determine ACs' roles in neurotransmission and their synaptic connectivity (Catterall, 2011; Kaneda et al., 2007).

Over the past decade, transcriptomic studies have significantly expanded our understanding of AC diversity, revealing that the number of ACs in the mouse retina corresponds to twice what researchers thought (Fig. 1*B*). For example, the drop-seq technique, which allows thousands of individual cells to be profiled simultaneously at the single-cell level, provided a significant leap forward. Macosko et al. (2015) used this method to identify 21 distinct AC clusters, aligning with earlier predictions (Masland, 2012). However, in a further study, the same authors pointed out that the technique could not fully resolve all the retinal cell types within the different classes (Yan et al., 2020). What was the cause of this limitation? Eighty percent of retinal cells correspond to rod photoreceptors (Jeon et al., 1998), making the ACs that are less abundant often undersampled. This under-representation made capturing the rarest types or resolving subtypes with similar gene expression profiles challenging. As a workaround, Yan et al. (2020) developed a selective depletion strategy to tackle this limitation. Transgenic mice and fluorophore-conjugated antibodies were used to label all non-AC cells, and they effectively filtered them out before performing droplet-based scRNA-seq. This technique analysed only cells positive for key markers like the transcription factor *Pax6* and the vesicular inhibitory amino acid transporter *Slc32a1*. With this approach, the researchers could profile 32,523 ACs from the mouse retina, and using an unsupervised classification they uncovered 63 distinct AC types. By comparing expression patterns of well-known type-specific markers, they could assign clusters to established AC types such as AII, starburst, A17 and VIP (see Table 1). However, some ACs, like TH2 ACs, could not be matched to any known markers. A similar approach was used recently, generating the largest retinal scRNA-seq dataset (190,000 single cells), identifying 67 AC types (Li et al., 2024), and showing that four subtypes were under-clustered in Yan et al. (2020).

Interestingly, fewer AC clusters have been identified in primates, with 27 clusters in the fovea and 34 in the periphery of the macaque retina (Peng et al., 2019), as opposed to approximately 60 in the mouse retina. This disparity could be due to sample size limitations, as early studies of mouse retinas showed that analysing fewer cells often resulted in fewer identified subtypes, suggesting that AC diversity in primates may have been similarly underestimated. Alternatively, a teleological explanation suggests that species such as mice, chicks and fish rely more on retinal processing, while primates depend more on cortical processing (Shekhar & Sanes, 2021).

The transcriptome-based classification mentioned above provides a comprehensive atlas for targeting AC types and establishes a standardized reference across laboratories. Its relevance is underscored by recent findings that retinal molecular architecture is highly conserved across 17 vertebrate species (Hahn et al., 2023); it could be that specific molecular markers can be used across species to target homologous cell subclasses.

When molecular signatures can be aligned with morphological traits – such as dendritic tree size, lamination within the IPL, and connectivity with bipolar cells and GCs – it becomes possible to classify AC types. However, in most cases, it remains challenging to establish a clear correspondence between morphology and genetic clusters. For example, as previously mentioned, Helmstaedter et al. (2013) identified 44 cell types using morphological criteria (Fig. 1*C*), but only a subset has been successfully matched to molecular types, including OFF and ON starburst ACs, AII ACs, VIP-1 and VIP-2 ACs, A17 ACs, as well as SEG and sign-inverted (SI) ACs (Fig. 1*C*). To provide a comprehensive summary, we compiled data from existing literature on AC light responses, morphological characteristics and transcriptomic clusters into a single table. Through this analysis, we identified 18 distinct AC types characterized by at least one of these levels. Among those, six AC types exhibit subtypes: CRH (3 subtypes), nNOS (2), VIP (3), Somatostatin (SST) (2), and the non-GABAergic non-Glycinergic (nGnG) (4), resulting in a total of 27 different ACs being studied out of the 67 ACs described through transcriptomic, assuming no overlap. The AC subtype positive to the growth hormone-releasing hormone (GHRH AC) has been solely identified through single-cell transcriptomic studies (Yan et al., 2020) and not in terms of morphology and physiology. The remaining ACs in Table 1 are relatively well-characterized based on key features such as neurotransmitters, polarity, dendritic field size, stratification, and soma localization. However, the neurotransmitter identity is still unknown for some groups, like the nGnG ACs. Despite this progress, the direct alignment between molecular classifications and morphological studies remains limited.

This comprehensive classification highlights the necessity of integrative studies that combine molecular profiles, detailed morphological reconstructions and connectivity analyses to resolve the full diversity of ACs. Despite significant progress, many AC types remain partially characterized, with gaps in linking transcriptomic clusters to defined structural and functional properties. Future research should address these gaps systematically by identifying the neurotransmitter types of unclassified ACs, improving the definition of subtypes, and exploring how morphological and molecular diversity influences visual processing.

*Genetic tools and mouse models for studying the different types of ACs.* Molecular studies have been instrumental in identifying genes with cell-type-specific expression patterns, enabling targeted investigations of AC subtypes. A library of 230 promoters has been developed to target various retinal cell types, with 62 showing specificity for ACs (Jüttner et al., 2019). However, most of these promoters remain uncharacterized regarding their exact AC targets. These advances, combined with the development of sophisticated genetic tools such as Cre-driver lines, knockout (KO) models and reporter lines, have expanded the ability to study AC diversity at unprecedented levels of precision. Together, these tools facilitate the exploration of previously inaccessible AC types and open new avenues for understanding their contributions to retinal circuitry and visual processing.

*Cre-lox mice lines for AC targeting.* One of the most effective genetic tools developed for studying specific cell types, including retinal ACs, is the Cre-LoxP system (Madisen et al., 2010). This tool allows for selective activation or deletion of genes in specific cell types. It works by using Cre recombinase, which is expressed under cell-type-specific promoters to target and modify genes only in those cells. This can be combined with reporter lines that express fluorescent proteins, facilitating visualization and tracking of ACs and making it possible to study their morphology, connectivity and responses to visual stimuli.

Mice lines expressing Cre recombinase under AC-specific promoters have been developed, enabling targeted genetic manipulation of these cells for functional studies. It is possible to target distinct ACs based on their molecular properties by using promoters tied to the expression of neurotransmitters (e.g. GABA, glycine, glutamate) or neuropeptides (e.g. VIP). For instance, some Cre-lines allow a broad study of ACs by targeting the whole population of ACs that are GABAergic (Gad2-Cre), glycinergic (GlyT2-Cre) or wide-field ACs (Eulenburg et al., 2018; Korympidou et al., 2024; Lei et al., 2024; Park et al., 2024). However, one caveat is that they do not allow separating between specific cell types, and some off-targets can also be found (Berry et al., 2023; Eulenburg et al., 2018; Vuong et al., 2015).

It is also possible to study a more specific subtype by targeting its specific peptide, enzymes, or transporter, such as the ChAT-Cre that targets the starburst ACs (Pottackal et al., 2020), Crh-Cre (corticotropin-releasing hormone-Cre) (Jacoby et al., 2015; Park et al., 2018; Zhu et al., 2014), nNOS-Cre (neuronal nitric oxide synthase-Cre) (Park et al., 2020), VGluT3-Cre (vesicular glutamate transporter 3) (Hsiang et al., 2017; Lee et al., 2016) and VIP-Cre that target a subset of inhibitory ACs (Zhu et al., 2014). Recently, a novel intersectional approach also combined, elegantly, the Cre and Flpo

recombinases system to target a subset of non-spiking wide-field ACs (Lei et al., 2024; Park et al., 2024). They combine two genetic tools, Cre and Flpo recombinases, with a reporter to label Bhlhe22$^+$ cells, which targets GABAergic ACs. By crossing *Bhlhe22*-Flpo and the kappa opioid receptor (KOR)-Cre lines, they selectively label some ACs, which they named B/K ACs. We have summarized all the mouse lines targeting different AC types in Table 1.

*Knockout mice line for ACs targeting.* In addition to Cre lines, conditional knockout (KO) models have been invaluable for studying the roles of specific genes in AC development and function. For example, VGlut3 KO mice (VGluT$^{-/-}$) show how the neurotransmission from ACs contributes to contrast sensitivity and motion detection (Kim et al., 2015). One limitation of knockout mouse lines is the potential for compensatory mechanisms to arise in neural circuits (Nelson & Valakh, 2015). To address this, it is also possible to selectively eliminate specific ACs in the adult stage by inducing the expression of the diphtheria toxin receptor through Cre-mediated recombination. For example, using this approach, Tien et al. (2016) demonstrated the role of vGluT3 ACs in shaping SbC-GC responses in a contrast- and size-selective manner.

Such models allow researchers to explore gene function at specific developmental stages or under physiological conditions, thereby providing insights into the molecular mechanisms underlying AC diversity and function. Other KO models used to study ACs are the *Prdm13* KO, which targets a subset of GABAergic and glycinergic ACs (Watanabe et al., 2015); the *Slc32a1* KO, which eliminates the GABA transporter in starburst ACs (Chen et al., 2016); and the *Gabra2* KO, which eliminates the $\alpha 2$ subunit of GABA-A receptors in starburst ACs (Chen et al., 2016).

Advances in genetic targeting are transforming research on retinal ACs, enabling the precise study of their diversity and function. Similar to how genetic tools revolutionized GC research (Kerschensteiner, 2022), they are now poised to uncover novel aspects of ACs. However, one unresolved issue remains that each subtype may have a different expression pattern across the visual field. This is the case of GC types (Bleckert et al., 2014; Hughes et al., 2013; Kim et al., 2008). Future studies incorporating spatial transcriptomics and refined mouse models will be key to comprehensively mapping AC diversity and function.

## How do amacrine cells respond to light?

As discussed in 'Diversity in morphology of amacrine cells', understanding how ACs respond to light begins with examining their anatomy, which provides crucial insights into their functional roles. For instance, mono-stratified ACs exhibit polarity based on their inputs, determining their ON or OFF selectivity. However,

identifying their precise pre- and postsynaptic partners remains a significant challenge. Wide-field ACs are generally thought to integrate inputs over larger areas and have broader receptive fields than narrow-field ACs, though this is not always the case. The presence of surround suppression – already evident in these cells – further complicates this relationship.

Additionally, the link between input stratification and physiological function is far from straightforward (Jo, Deniz, Xu et al., 2023). This complexity is further illustrated by the compartmentalization observed in many AC subtypes and the intricate stimulus selectivity exhibited by certain ACs, such as starburst and VGluT3-expressing ACs (e.g. Kim et al., 2020; Park et al., 2024). Therefore, there is a clear need to investigate the functional selectivity of ACs at the compartmental level and to employ ecologically relevant stimuli to better understand their diverse roles in retinal signal processing.

**Stratification of amacrine cells.** A key insight into how ACs respond to light is to look at their anatomy. ACs form synaptic connections with bipolar cells and GCs, whose projections also reside within the IPL. Examining the positioning of an AC's dendrites provides clues about its interactions with retinal cells and its role in visual processing. ACs that project into the ON sub-lamina modulate signals related to light increments by connecting with ON cells. In contrast, those in the OFF sublamina interact with OFF cells to influence light decrement signals. Since monostratified ACs confine their dendrites in a single IPL layer, they specialize in processing information within specific visual circuits. For instance, CRH-1 ACs, a monostratified ON sub-type, suppress responses only to positive contrast in suppressed-by-contrast GCs (Jacoby et al., 2015). Another example is the monostratified VIP-3 ACs, which exhibit the most straightforward receptive fields among the three subtypes of VIP ACs and are characterized by spatially tuned ON responses (Akrouh & Kerschensteiner, 2015; Park et al., 2015). Bistratified ACs integrate signals from both the ON and OFF pathways (Masland & Raviola, 2000). Finally, pluristratified ACs modulate visual signals across multiple pathways with their broad dendritic stratification.

However, the relation between anatomical stratification and physiology is far from straightforward. The recent discovery of the sign-inverted AC type (SI-AC) challenges the assumption that stratification predicts ON/OFF selectivity. SI-ACs are narrow ACs that stratify in the ON layers but hyperpolarize in response to light spots. This unusual behaviour is due to their synaptic inputs: SI-ACs receive inhibitory signals from ON bipolar cells via the metabotropic glutamate receptor mGluR8 while

also receiving excitatory input through electrical synapses from an OFF wide-field AC (Jo, Deniz, Xu et al., 2023).

Given this complexity, directly recording AC responses to light is essential for understanding their role in visual processing. Table 1 summarizes known AC light responses.

**Size of the dendritic tree of amacrine cells.** The size of an AC's dendritic tree plays a crucial role in processing visual information by determining how these cells receive and integrate synaptic input. Their complex branching and stratification within retinal layers allow them to modulate light-driven signals and shape visual responses.

Narrow-field ACs have compact dendritic trees that enable them to process information locally (Fig. 1) and may help distinguish luminance from contrast (Werblin, 2010). Some, like AII ACs, sharpen temporal signals by generating fast, regenerative currents in response to light. On the other hand, wide-field ACs, with their extensive dendritic trees, integrate visual information across large retinal areas, contributing to global visual processing.

However, the relationship between dendritic tree size and receptive field properties is more complex than it may seem. In GCs, for instance, surround suppression can cause the receptive field centre to be smaller than the dendritic tree (Gauthier et al., 2009). A similar phenomenon occurs in ACs, where many types show reduced responses to large stimuli. This is the case of AII ACs, where this suppression is mediated by NOS-1 ACs (Nath et al., 2023; Park et al., 2020; Völgyi et al., 2002). Other AC subtypes, including VIP$^+$ ACs (Akrouh & Kerschensteiner, 2015), starburst ACs (Chen et al., 2020; Lee & Zhou, 2006) and VGluT3 ACs (Chen et al., 2017) also exhibit surround suppression.

**Cellular compartmentalization within amacrine cells.** Compartment-specific processing of synaptic signals between the soma and dendrites is fundamental to neural computations in the central nervous system (Acarón Ledesma et al., 2024; Branco & Häusser, 2010; Stuart & Spruston, 2015). In the retina, ACs exhibit remarkable diversity in processing and integrating visual signals, where dendrites often shape the input–output relationships with the postsynaptic partner.

In most cases, the same AC's dendritic branch receives the inputs and generates outputs, allowing the information to bypass the soma. This decentralized processing approach is believed to optimize the use of cellular resources.

ACs can integrate the visual signal from their dendrites in a global or local strategy. For instance, proximal dendrites of AII ACs locally collect excitatory glutamatergic inputs from rod bipolar cells and inhibitory inputs from ACs (Demb & Singer, 2012). In contrast, each dendrite of starburst ACs independently detects motion in a specific direction (Euler et al., 2002). This directional selectivity arises from a combination of dendritic morphology, synaptic inputs, and the passive and active membrane properties that shape the centrifugal preference of starburst AC dendrites (for an in-depth discussion of different models of centrifugal direction selectivity in starburst ACs, see Wei, 2018). This complex strategy allows the starburst ACs to selectively inhibit signals from one direction while facilitating signals from the opposite direction (Fig. 2*F*).

However, the electronic isolation of different compartments is not a trivial matter, and is a consequence of several factors that can leave starburst ACs comprising >20 functionally distinct compartments (Poleg-Polsky et al., 2018). When motion signals activate synaptic inputs on starburst AC dendrites, the metabotropic glutamate receptor 2 regulates the threshold for the initiation of calcium transients in dendrites to generate the direction selectivity (Koren et al., 2017), while the soma acts as a low-pass filter to prevent individual dendrites from being affected by the directional responses of other dendritic sections. This mechanism allows each dendritic segment to function independently. A recent study has identified that perisomatic Kv3 channels are the key regulators in stabilizing the membrane potential around the soma of starburst ACs, providing this voltage-dependent shunt (Acarón Ledesma et al., 2024). Additionally, inhibitory mechanisms contribute to compartmentalization: blocking GABAergic input reduces functional compartmentalization, suggesting that inhibition is essential in maintaining independent dendritic processing (Koren et al., 2017; Poleg-Polsky et al., 2018). Remarkably, this inhibition differs between ON and OFF starburst ACs (Chen et al., 2016).

Another great example of compartmentalization is the A17 AC, which participates in the rod pathway and gives reciprocal feedback to rod bipolar cells (Chávez et al., 2006; Dong & Hare, 2003). These cells deliver reciprocal feedback inhibition to presynaptic bipolar cells through hundreds of independent microcircuits operating in parallel (Grimes et al., 2010). This can help the circuit isolate feedback microcircuits and maximize its capacity to handle many independent processes.

Compartmentalization is a fundamental principle in understanding the diverse and specialized roles of ACs in retinal circuits. By enabling highly localized processing, a single cell can execute multiple parallel computations, such as direction selectivity and independent dendritic signal integration (Chen et al., 2017). Despite significant progress, the mechanisms governing compartmentalization – particularly its functional diversity across different AC types – remain incompletely understood. Advances in recording techniques, such as voltage and GABA imaging, are

promising for further elucidating how ACs process light at the compartmental level.

While patch-clamp techniques remain a powerful tool, they are limited to somatic recordings. In contrast, calcium imaging has provided a broader perspective, enabling the monitoring of activity across AC dendrites and uncovering compartmentalized responses in certain types (Euler et al., 2002; Korympidou et al., 2024; Poleg-Polsky et al., 2018). Voltage imaging, though still in its early stages, shows great promise for capturing rapid, localized changes in membrane potential across AC processes (Acarón Ledesma et al., 2024). Additionally, the potential for GABA imaging could open new avenues for studying inhibitory signalling directly (Matsumoto et al., 2024). Collectively, these technological advancements are set to drive significant progress in unravelling the diverse and essential roles of ACs in visual processing.

**More complex types of selectivity.** Dendritic recordings have revealed complex stimulus selectivity in ACs. As mentioned earlier, one of the best-known examples is the starburst AC, whose dendrites exhibit selective responses to centrifugal motion (Euler et al., 2002). Another example is VGlut3 AC, whose dendrites are selective to approaching motion (Kim et al., 2020) and to object motion (Hsiang et al., 2017). A recent study has also shown that a group of GABAergic wide-field ACs display orientation-selectivity in their dendrites, providing orientation-tuned inhibition in non-orientation-tuned GCs and bipolar cells (Lei et al., 2024; Park et al., 2024).

To better understand AC response to light and visual function, examining functional selectivity at the dendritic compartment level and incorporating more naturalistic stimuli is crucial. This approach has already transformed our understanding of GCs (Gollisch & Meister, 2010). Currently, most studies rely on simple visual stimuli when investigating AC function. However, research using more naturalistic stimuli (Chen et al., 2016; Kim et al., 2015) suggests that many AC subtypes perform intricate computations within individual compartments that remain largely unexplored. Connectomics might help constrain the network models necessary to explain these results.

## How do retinal amacrine cells influence ganglion cell function?

ACs play a crucial role in shaping the way GCs process visual information. By forming synaptic connections with both GCs and bipolar cells, ACs contribute to key computations such as motion detection, approaching motion, and direction selectivity. These functions are mediated through complex networks of inhibitory and excitatory synapses. In this section, we explore the synaptic partners of ACs, the structural diversity of their synapses, and the functional circuits that allow ACs to influence GC activity.

### Amacrine cells' postsynaptic partners

*Diversity in postsynaptic partners.* The number and type of postsynaptic partners that ACs can contact vary depending on the AC type itself (Helmstaedter et al., 2013) (Table 2). Some ACs establish connections with a wide variety of cells. For example, AII ACs interact with four different partners: ON and OFF cone bipolar cells, OFF GCs, AII cells, and the NOS-1 AC (Famiglietti Jr & Kolb, 1975; Marc et al., 2014; Park et al., 2020) (Table 2 and Fig. 2*E*). Another example is the dopaminergic ACs, which contact AII ACs, other ACs, and horizontal cells as well (Contini & Raviola, 2003; Herrmann et al., 2011; Kolb et al., 1991; Pourcho, 1982). VGluT3 ACs also contact a variety of functionally distinct GCs (Famiglietti Jr & Kim et al., 2015; Kolb, 1975; Lee et al., 2014, 2016).

Conversely, some other ACs have highly specific synaptic targets. This is the case of A17 ACs, which receive input exclusively from rod bipolar cells and provide output to the same cell type (Grimes et al., 2015; Nelson & Kolb, 1985). However, for many ACs, their precise synaptic partners remain unidentified. A summary of known AC connectivity is provided in Table 2.

The retina's layered architecture ensures that each AC type has a distinct set of input and output partners, which shape its functional properties (Sanes & Zipursky, 2010). For instance, monostratified medium-field starburst ACs, which are crucial for detecting directional selectivity, specifically target direction-selective GCs within the layer where these GCs extend their dendrites (Briggman et al., 2011; Ding et al., 2016) (Fig. 2*F*; Famiglietti, 1991). By forming synaptic connections with other ACs, bipolar cells and GCs, ACs regulate visual processing through a combination of inhibitory and excitatory mechanisms. These interactions shape visual computations, driven by diverse chemical and electrical signalling pathways.

### Amacrine cell synapses

*Chemical synapses of amacrine cells.* ACs use a wide range of neurotransmitters and neuromodulators to communicate with GCs and other retinal neurons. While their primary neurotransmitters are inhibitory, such as GABA and glycine, some ACs also release excitatory neurotransmitters, including glutamate, acetylcholine and dopamine. This versatility enables ACs to contribute to complex networks that fine-tune visual signals.

*Inhibitory neurotransmitters used by amacrine cells.* Most ACs inhibit their target neurons using either GABA or glycine. As previously mentioned, wide-field ACs are predominantly GABAergic, while narrow-field ACs are mainly glycinergic, though some also release GABA (see

**Table 2. Main inputs and outputs of known amacrine cell types**

| AC type | Subtypes | Inputs | Outputs | Role |
|---|---|---|---|---|
| AII (Marc et al., 2014; Demb & Singer, 2012) | —— | RBC, AII | On Rbc, Off Rbc, Aii | Cross-over inhibition from the ON pathway to the OFF pathway; feedforward excitation |
| Starburst (Wei, 2018) | — | On and OFF RBC (depending on subtype) | DS Rgcs | Direction selective |
| A17 (Grimes et al., 2010) | — | RBC | RBC, AII | Feedback inhibition to improve the signal-to-noise ratio |
| VGluT3 (Tien et al., 2016) | — | BC (type 3 and 5), other ACs | W3 RGC, SbC RGC | Motion sensitive |
| TH2 (Knop et al., 2011) | ? | Type 3 OFF, type 5 ON BCs. | W3 RGC, AII, HD2p-RGC | Differential response to global and local motion |
| CRH (Park et al., 2018) | CRH-1 | ON BCs | ON alpha GC and Sbc GC | Mediates contrast sensitivity |
|  | CRH-2 |  | ? | ? |
|  | CRH-3 |  | ON alpha ganglion cell | Stabilize ON alpha GC with tonic inhibition in background light |
| nNOS | nNOS1 | ON BCs | AII, RBC | Provide inhibitory surround to AIIs under mesopic conditions |
|  | nNOS2 | ON and OFF BCs | nNOS-2 ACs | Reduce conductance of gap junctions |
| VIP (Park et al., 2015; Akrouh & Kerschensteiner, 2015) | VIP-1 | ON and OFF BCs (types 1 and 2) | ∼ OFF delta GC, W3 GC | ∼ Play a role in contrast sensitivity and motion detection |
|  | VIP2 | ∼ ON BCs (types 5, XBC, 6, 7 and 8) | ∼ W3 GC, ON-OFF DS | ∼ Disinhibition mechanism reducing tonic inhibition |
|  | VIP-3 | ∼ ON BCs (types 5, XBC, 6, 7 and 8) | ? | ∼ Create inhibitory subunits that could modulate GC activity |
| SST (Zhu et al., 2014) | SST-1 ? | ∼ ON and OFF BCs | ? | ? |
| B/K (Park et al., 2024) | ? | ∼ ON and OFF BCs | OFF Delta RGC and ON Alpha RGC | ? |
| nGnG | nGnG-1 | ? | ? | ? |
|  | nGnG-2 | ? | ? | ? |
|  | nGnG-3 | ? | ? | ? |
|  | nGnG-4 | ? | ? | ? |
| SEG | ? | ? | ? | ? |
| GHRH | ? | ? | ? | ? |
| PENK | ? | ? | ? | ? |
| SRIF (Vuong et al., 2015) | ? | ? | ∼ Dopaminergic ACs and M1 ipRGCs | ? |
| S3-Gbx2+ (Kerstein et al., 2020) | ? | ON and OFF BCs | ∼ W3B RGCs | ∼ Object motion detection |
| SI-AC (Jo, Deniz, Xu et al., 2023) | ? | ON BCs, ∼ bistratified wide-filed AC | AIIs, VGlut3-ACs, ON-RGCs, and ON-OFF RGC | Local crossover inhibition |
| COMS-AC (Jo, Deniz, Cherian et al., 2023) | ? | OFF BCs, GABAergic ACs | HD2p-RGC | Local motion sensitivity |
| PAS4/5 (Jia et al., 2020) | PAS4/5-1 | vGlut3 AC ACs | ? | Local contrast |
|  | PAS4/5-2 | vGlut3 AC ACs | ? | Local contrast |

The references used to create this table are listed under each AC name. A question mark (?) is added if no information was found. If only a hypothesis has been proposed for the corresponding connectivity or function, a tilde symbol '∼' is added in front of the sentence.

Table 1) (Chen et al., 2011; Menger et al., 1998; Pourcho & Goebel, 1985). This division into glycinergic and GABAergic ACs allows specific inhibitory effects, which filter the visual signals transmitted to GCs.

GABA is the most common inhibitory neurotransmitter used by ACs in the retina, with more than half of ACs classified as GABAergic (Korympidou et al., 2024; Matsumoto et al., 2024; Yan et al., 2020). Although both GABAergic and glycinergic receptors are evenly distributed throughout the IPL, different receptor subtypes contribute to distinct inhibitory functions. The retina expresses multiple GABA receptor types (GABA-A, -B, and -C), as well as glycinergic receptors, and specific GCs preferentially express particular subtypes (Gustincich et al., 1999; Koulen et al., 1996). This receptor specificity allows ACs to selectively inhibit certain GC types by sharpening their spatial tuning. For example, transient ACs, which respond to rapid changes in light intensity, provide inhibitory input to transient GCs, allowing them to shape their responses to fast-moving stimuli (Masland, 2012).

*Excitatory neurotransmitters used by amacrine cells.* Although ACs are primarily inhibitory interneurons, they can also influence GCs through excitatory neurotransmitters such as glutamate and acetylcholine. Whereas it is less common than GABA and glycine, glutamate is used by some ACs. It is an excitatory neurotransmitter primarily released by photoreceptors and bipolar cells. These ACs, identified by their immunoreactivity to VGluT3, are known as VGluT3 ACs or glutamatergic ACs (GACs) (Lee et al., 2014, 2016). By releasing glutamate onto other retinal neurons, VGluT3 ACs create feed-forward excitatory circuits that interact with inhibitory networks (Jia et al., 2020). These VGluT3 ACs can modulate both excitatory and inhibitory pathways, using glutamate and glycine, respectively (Chen et al., 2017; Grimes et al., 2011; Hsiang et al., 2017; Jia et al., 2020; Lee et al., 2014, 2016). Behavioural studies have also linked VGluT3 ACs with threat responses (Kim et al., 2020).

Acetylcholine is another key excitatory neurotransmitter released by starburst ACs (Masland & Livingstone, 1976). Starburst ACs specifically release this cholinergic signalling onto direction-selective GCs, which respond to motion in specific directions, as these GCs express nicotinic receptors, which are the target of acetylcholine (Zucker & Yazulla, 1982), making them respond to motion in specific directions. While the exact role of acetylcholine is not yet fully understood, it is shown to be critical in mediating direction-selective responses in the retina, controlling the timing of the direction-selectivity GC excitatory responses (Masland & Cassidy, 1987; Vaney et al., 2012; Wei, 2018), through dendro-dendritic

synapses (Broma's et al., 2017), especially in low contrast conditions (Sethuramanujam et al., 2016).

*Neuromodulators used by amacrine cells.* Beyond neurotransmitters, ACs also release a variety of neuromodulators, including dopamine, endorphins (Gallagher et al., 2010), somatostatin (Cervia et al., 2008), neuropeptide Y (D'Angelo & Brecha, 2004), endocannabinoids (Estay et al., 2024; Vielma et al., 2020; Yazulla, 2008), nitric oxide (Hoffpauir et al., 2006; Vaney, 2004; Vielma et al., 2012; Wilson et al., 2011), substance P (Moya-Díaz et al., 2024) and VIP (Casini & Brecha, 1991). These neuromodulators act more slowly than neurotransmitters and diffuse across a broader retinal area. This diverse signalling allows ACs to precisely regulate visual processing across multiple circuits, enabling them to perform specialized functions within the retina. For example, dopaminergic ACs release dopamine in response to light to shift the balance between rod and cone pathways, which modulate their sensitivity and responsiveness, to optimize visual processing for different lighting conditions (Popova, 2014). Thanks to this mechanism, dopamine ACs influence signal transmission across circuits that adjust between scotopic and photopic vision (Masland, 1988)

*Amacrine cell co-release of neurotransmitters and neuromodulators.* An essential feature of ACs is that many release more than one type of neurotransmitter (Hirasawa et al., 2012; Lee et al., 2016; O'Malley et al., 1992), suggesting that individual cells play multiple roles. This diversity of output enriches the palette with which ACs can modulate the activity of GCs.

While ACs seem never to co-release the two primary inhibitory neurotransmitters, GABA and glycine (Table 2), some types can release both an inhibitory and an excitatory transmitter. The most well-known example is the starburst ACs, which co-release GABA and acetylcholine (O'Malley et al., 1992). Starburst ACs release acetylcholine, which potentiates the response to movement in all directions, and GABA, which inhibits responses to stimuli moving in non-preferred directions to mediate direction selectivity (Vaney et al., 2012; Wei, 2018). This dual-transmitter release helps GCs to respond robustly to stimuli moving in their preferred direction (Demb, 2007; Marshak, 2016). Another example is the co-release of glycine and glutamate by VGluT3 ACs (Haverkamp & Wässle, 2004): while glycine inhibits uniformity detector GCs (Lee et al., 2016), glutamate excites GCs that are responsive to contrast and approaching motion (Lee et al., 2014, 2016; Tien et al., 2016), or GCs critical for behavioural responses (Kim et al., 2020).

Compared to ACs that release only one neurotransmitter, co-release allows for more precise temporal control, particularly in processing motion and contrast

changes. For instance, starburst ACs first release acetylcholine in the preferred direction, followed by GABA, whereas in the null direction, both neurotransmitters are released simultaneously (Sethuramanujam et al., 2016; Wei, 2018).

Some AC types can also release a neurotransmitter and a neuromodulator, e.g. dopaminergic ACs co-release dopamine and GABA (Hirasawa et al., 2012), which is likely to help the retina transition between different lighting conditions (Pérez-Fernández et al., 2019). Following light exposure, dopaminergic ACs release dopamine via volume transmission, adjusting retinal circuit gain for bright-light vision. Additionally, they form GABAergic synapses with AII ACs, where GABAergic input may establish an inhibitory level under bright conditions, preventing signals from saturated rods from entering the cone pathway, allowing the retina to adapt more smoothly to changes in brightness (Hirasawa et al., 2015).

The ability of ACs to co-release multiple neurotransmitters or a combination of neurotransmitters and neuromodulators highlights their versatility and complexity in retinal circuits. This feature enables ACs to perform diverse and specialized roles, from direction selectivity to adapting the retina to different light conditions. By integrating excitatory and inhibitory signals with precise temporal control, ACs contribute to the dynamic range and efficiency of retinal processing. While considerable progress has been made, there is still much to understand about the mechanisms and functional implications of neurotransmitter co-release in ACs. Advanced techniques such as optogenetics and high-resolution imaging (e.g. dopamine sensors) will be essential for uncovering how these dual-signalling systems influence retinal processing.

*Electrical synapses of amacrine cells.* In addition to chemical synapses, some ACs also communicate via electrical synapses through gap junctions (Bloomfield & Völgyi, 2009). These gap junctions allow the flow of ions and small molecules between cells, enabling rapid and synchronized signalling. Several types of connexins are expressed in the retina, each serving distinct functional roles, with the most common being Cx36, Cx45 and Cx50 (Schubert et al., 2005; Söhl et al., 2010; Völgyi et al., 2005).

ACs form gap junctions that connect with neighbouring cells, such as other ACs, bipolar cells, and GCs (Pan et al., 2010). These gap junctions are essential for regulating retinal responses to changes in light intensity. The strength of gap junction coupling between ACs can be dynamically modulated in response to environmental stimuli, such as varying light levels (Curti & O'Brien, 2016). Depending on ambient light conditions, ACs adjust their coupling to regulate retinal responses to different light intensities and contrasts. This regulation helps the retina maintain sensitivity across a wide range of light levels, ensuring efficient visual function and adaptation to changing lighting conditions.

Gap junctions also enable rapid synchronization of electrical activity across large networks of cells. All ACs, which are part of the rod pathway, are activated by rod bipolar cells and are critical for transmitting rod-driven information during scotopic vision. These cells are electrically coupled to one another and to cone bipolar cells, forming a network that efficiently relays rod-driven signals to GCs (Bloomfield & Völgyi, 2009; Demb & Singer, 2012). However, a key disadvantage of electrical coupling is its impact on spatial resolution. Because AII ACs are strongly coupled to one another and to cone bipolar cells, electrical signals from individual photoreceptors can spread laterally across the network. While this lateral spread enhances sensitivity under low-light conditions by integrating dim light signals, it also causes spatial blurring as fine visual details are averaged out. This trade-off is particularly significant in scotopic conditions, where electrical coupling is higher than in photopic conditions (Dunn et al., 2006; Vardi & Smith, 1996). In low-light situations, maximizing sensitivity and reducing noise take precedence, even if spatial detail is compromised. In contrast, during bright-light conditions, reduced electrical coupling enhances spatial resolution, preserving fine visual details. In conclusion, the retina adjusts gap junction coupling based on ambient light levels to strike a balance between sensitivity and resolution. Interestingly, the strongest coupling is observed under intermediate light conditions, optimizing the signal-to-noise ratio and facilitating luminance adaptation (Bloomfield & Völgyi, 2009). This plasticity is also influenced by factors like dopamine levels and circadian rhythms, which regulate connexin phosphorylation and dephosphorylation (Bloomfield & Völgyi, 2009).

Many ACs use both inhibitory chemical synapses and electrical gap junctions. As a result, if these gap junctions transmit enough electrical charge, an AC can simultaneously be excitatory at some synapses and inhibitory at others. A hypothesis for the functional role of this excitatory–inhibitory balance is that gap junctions provide correlated signals that reflect the global properties of the visual scene (light, contrast, global object). In contrast, the signal output given by the neurotransmitters' release gives local information about the stimulus (edges, details) (Trenholm et al., 2014). The two signals are independent but complementary. This type of double connectivity is also essential for synchronizing the activity of AC among the same network and facilitating the spread of inhibitory signals over large areas of the retina (Masland & Raviola, 2000). The combined inhibitory and excitatory signalling through gap junctions has been described in various AC types, including the AII AC (Hartveit & Veruki

2012), the A17 ACs (Elgueta et al., 2018), the A8 ACs (Lee et al., 2015; Yadav et al., 2019), the nNOS2 ACs (Jacoby et al., 2018; Zhu et al., 2014), and the VIP ACs (Park et al., 2015).

**Functional circuits of amacrine cells.** The anatomy and synaptic connectivity of ACs are intricately linked with their visual functions. Features like their dendritic arborization size and laminar positioning partially help determine the function of AC subtypes. At the same time, their synaptic connections enable them to modulate visual signals at both local and global levels. Together, these properties allow ACs to form various circuits that refine the visual signal in a variety of ways (Fig. 2).

Some ACs are highly specialized, interacting with specific GC types or circuits, whereas others are versatile, participating in multiple pathways and contributing to diverse visual computations. Developing cutting-edge genetic tools, such as Cre lines and knockout models, has revolutionized our ability to dissect these intricate circuits. These developments provide promising avenues for linking specific AC types to their distinct roles in visual processing. The key circuits involving ACs and GCs include lateral inhibition, disinhibition, feed-forward inhibition, crossover inhibition and excitation, and electrical coupling through gap junctions. A summary of these interactions is shown in Fig. 2.

*Surround suppression.* Surround suppression is a process where the stimulation of the peripheral regions of a receptive field actively inhibits the centre's response (Barlow, 1953; Buldyrev & Taylor, 2013; Flores-Herr et al., 2001; Ichinose & Lukasiewicz, 2005; Jacoby & Schwartz, 2017; Kuffler, 1953; Zhang et al., 2012). Classic experiments show that larger visual stimuli reduce retinal responses compared to smaller ones (Cook & McReynolds, 1998; Jacoby & Schwartz, 2017). This mechanism, often called lateral inhibition, extends beyond the retina and is observed in other brain regions as well (Fisher et al., 2017; Ozeki et al., 2009; Usrey & Alitto, 2015).

In the retina, surround suppression is primarily mediated by horizontal cells and ACs, particularly wide-field ACs, where the level of suppression is influenced by ambient light conditions (Baccus et al., 2008; Famiglietti, 1992; Farrow et al., 2013; Nath et al., 2023; Park et al., 2020). This inhibition underpins the classic 'centre–surround' antagonistic organization of GC receptive fields, where the centre and surround regions respond oppositely to light (Fig. 2*A*) (Barlow, 1953; Kuffler, 1953). For example, an ON-centre receptive field is paired with an OFF-surround, and vice versa. Additionally, ON cells often exhibit OFF-surround activation, leading to depolarizing responses to dark stimuli in the periphery (Barlow, 1953; Kuffler, 1953),

while OFF cells exhibit a similar surround antagonism but with opposite polarity.

Surround suppression can be inherited from bipolar cells, either through suppression mediated by horizontal cells (Hartline & Ratliff, 1957; Hartline et al., 1956), or via AC-driven inhibition at the terminals of bipolar cells and ACs (Borghuis et al., 2013; Franke et al., 2017; Gaynes et al., 2022; Roy et al., 2024; Strauss et al., 2022; Swygart et al., 2024). For instance, Park et al. (2020) combined functional imaging and connectomics to demonstrate that nNOS-1 wide-field ACs constitute a significant source of inhibition in the rod bipolar cell pathway, forming synapses with both AII ACs and rod bipolar cells. Likewise, Nath et al. (2023) showed that nNOS-1 ACs also provide inhibitory surround modulation to AII ACs under mesopic conditions, with this modulation propagating downstream to influence s-ON alpha GC responses.

Yet, surround suppression can also emerge independently in GCs (Buldyrev & Taylor, 2013; Farrow et al., 2013; Jacoby & Schwartz, 2017; Roska et al., 2000). A significant challenge in studying this phenomenon is distinguishing the contributions of horizontal cells and ACs (Ichinose & Lukasiewicz, 2005). Approaches such as genetic targeting and pharmacology have helped clarify the role of specific AC types in this process. For example, Johnson et al. (2018) genetically targeted pixel-encoder GCs in mice and demonstrated that this GC type receives exclusively inhibitory input from the surround. Using whole-cell patch-clamp recordings in pixel-encoder GCs, in combination with pharmacology, they further identified that spiking GABAergic ACs mediate this inhibition.

This mechanism also reduces redundancy by preventing neighbouring GCs from responding to redundant stimuli (Barlow et al., 1957; Bisti et al., 1977; Cook & McReynolds, 1998; Franke et al., 2017; Muller & Dacheux, 1997). In some cases, some GCs display such a strong surround inhibition that they can completely suppress their response to large stimuli, making them highly sensitive to edges (van Wyk et al., 2006). In other cases, strong surround suppression makes GCs, such as W3 GCs, sensitive to small, local motion, allowing them to detect small moving objects, including overhead predators (Zhang et al., 2012).

*Disinhibition.* ACs not only provide direct inhibition but also inhibit other inhibitory ACs (GABAergic or glycinergic) through a process known as 'serial inhibition' (Deny et al., 2017; Eggers & Lukasiewicz, 2006, 2010; Roska et al., 1998; Zhang et al., 1997). In this mechanism, an AC inhibits other inhibitory ACs, thereby disinhibiting the target bipolar cell or GC, allowing it to fire in response to visual stimuli (Fig. 2*B*) (Eggers & Lukasiewicz, 2010; O'Brien et al., 2003).

The impact of serial inhibition on spatial processing – how the retina computes different sizes of visual stimuli – is poorly understood. Direct inhibition tends to be more spatially focused, whereas serial inhibition has a broader reach, allowing different sizes of light stimuli to engage these circuits selectively. Research suggests that larger stimuli activate a broader AC network, leading to serial inhibition and modulating bipolar cell sensitivity (Eggers & Lukasiewicz, 2010). This organization reveals that this disinhibitory circuit acts as a spatial filter, shaping the spatial sensitivity of bipolar cells to different-sized stimuli.

Disinhibition also plays a major role in driving excitatory responses in OFF GCs by regulating their sensitivity to light decrements (Manookin et al., 2008; Margulis & Detwiler, 2007; van Wyk et al., 2009). When light decreases, ON bipolar cells hyperpolarize, leading to a reduction in excitatory input to AII ACs, which in turn reduces their glycine release onto OFF GCs. This reduction in inhibition, or disinhibition, results in an inward current that enhances the response of OFF GCs. This effect is particularly strong at low contrast, contributing significantly to the overall response. However, as contrast increases, the role of disinhibition diminishes relative to direct excitation from OFF bipolar cells. This mechanism is essential for shaping the sensitivity of OFF GCs to visual stimuli, especially at low contrast, and ensures effective processing of contrast changes in the visual system.

Interestingly, serial inhibition plays a key role in the retina's ability to perform multiplexed computation – processing multiple visual features simultaneously. Deny et al. (2017) used large-scale recordings to reveal that fast OFF GCs encode two distinct aspects of a visual scene. Cells located near a moving object respond in a nearly linear fashion, encoding the object's position. Meanwhile, more distant cells are largely unaffected by the object's position and instead respond non-linearly to changes in its speed. Further experiments suggested that only a specialized disinhibitory retinal circuit – comprising two ACs – enables these distant cells to perform this speed-sensitive computation (Deny et al., 2017)

As we have discussed, disinhibitory motifs typically involve two inhibitory neurons in sequence, where inhibiting the second neuron results in an excitatory effect on the principal neuron. However, Chen et al. (2020) revealed an additional function of a disinhibitory microcircuit in the retina, challenging the conventional view that disinhibition simply reduces inhibition. Instead, their study demonstrated that serial inhibition within the starburst AC-direction selective GC microcircuit does not relieve inhibition but instead preserves it through an interaction between network dynamics and short-term synaptic plasticity. Their study showed that, under noisy visual conditions, starburst ACs receive inhibitory input from neighbouring starburst ACs. This suppression is essential because it prevents excessive activation of starburst ACs before a motion stimulus, thereby avoiding short-term synaptic depression at starburst AC–direction selective GC synapses. Without this regulation, starburst AC-mediated inhibition of direction-selective GCs becomes weaker due to synaptic depression, ultimately impairing direction selectivity.

Overall, disinhibitory circuits are still poorly understood, but improvements in targeted genetic manipulation and imaging of ACs might provide further insights into these circuits. For example, Franke et al. (2017) used the glutamate sensor iGluSnFR to record activity in over 13,000 bipolar cell axon terminals. Through pharmacological experiments, they revealed that glycinergic ACs primarily modulate bipolar cell output indirectly by inhibiting GABAergic ACs rather than directly on bipolar cells. These findings suggest that the disinhibitory role of ACs in retinal processing may be underestimated, highlighting the importance of exploring how prevalent and interconnected the amacrine cell network truly is.

*Feedforward inhibition.* Feedforward inhibition in the retina, driven by ACs, shapes the timing and precision of visual signals. In this circuit, ACs rapidly inhibit downstream cells, like GCs, in response to excitatory signals from bipolar cells (Fig. 2*C*) (Zhang et al., 1997). This fast targeted inhibition sharpens visual processing by shortening the response window (Lee et al., 2016; Tien et al., 2016; Werblin, 1970). By suppressing excessive excitation, feedforward inhibition of ACs ensures that GCs focus on the most important visual features, filtering out redundant information (Cafaro & Rieke, 2010; Demb & Singer, 2012).

This mechanism helps GCs respond to sudden changes in the visual scene, ensuring that GCs respond effectively to transient or rapidly shifting stimuli. By suppressing excessive excitation, feedforward inhibition allows GCs to focus on the most critical visual features (Lee et al., 2016; Tien et al., 2016; Werblin, 1970; Werblin & Copenhagen, 1974).

*Feedback inhibition.* Unlike feedforward inhibition, which can act downstream to inhibit neurons like GCs, feedback inhibition creates a self-regulating loop in which ACs inhibit the very cells that excite them, such as bipolar cells (Fig. 2*D*) (Dong & Werblin, 1998; Roska et al., 1998). This inhibition can be reciprocal or non-reciprocal (Burkhardt, 1972; Chávez et al., 2006; Chun et al., 1993; Dong & Werblin, 1998; Grimes et al., 2010). By inhibiting bipolar cells, ACs can reduce their output's strength and duration, ensuring that GCs are not overloaded with excessive excitation (Eggers & Lukasiewicz, 2010).

Feedback inhibition regulates visual processing by controlling the timing and intensity of signals passed to GCs (Eggers & Lukasiewicz, 2010; Völgyi et al., 2002),

controlling the gain of feedforward signals (Grimes et al., 2015), and improving the detection of contrast, edges and motion by preventing bipolar cells from sending continuous, unfiltered signals (Dong & Werblin, 1998; Euler & Masland, 2000). For example, GABAergic feedback at bipolar cell terminals regulates the dynamic range of bipolar cells (Euler & Masland, 2000) and their centre–surround receptive fields (Flores-Herr et al., 2001; Franke et al., 2017) and sharpens the timing of their responses (Dong & Hare, 2003). It has been proposed that this type of inhibition may play a role in how ACs contribute to the decorrelation of bipolar cell channels. This mechanism helps reduce the redundancy in the signals transmitted to GCs while preserving the functional diversity among the 15 distinct types of bipolar cells found in the mouse retina (Franke et al., 2017).

Feedback inhibition has also been suggested to indirectly modulate the surround responses of AII ACs under dark conditions (Völgyi et al., 2002). Although AII ACs exhibit minimal surround responses under scotopic conditions (Nath et al., 2023), it has been shown that feedback inhibition from an unidentified GABAergic AC onto the terminals of bipolar cells modulates this surround (Völgyi et al., 2002).

One classic example of feedback inhibition in the retina is the interaction between rod bipolar cells and A17 amacrine cells, where A17 ACs send reciprocal inhibitory signals to rod bipolar cell terminals (Chávez et al., 2006; Grimes et al., 2010). This feedback regulates the signal-to-noise ratio (or gain) of the rod bipolar cell–AII AC transmission (Grimes et al., 2015), indicating that the balance between excitation and inhibition provided by feedback inhibition optimizes the sensitivity of the rod pathway under dim light conditions.

*Crossover inhibition (or vertical inhibition).* Crossover inhibition allows inhibitory signals from one visual pathway to influence the opposite pathway. For instance, ACs can inhibit OFF bipolar cells or GCs when the ON pathway is active (Cafaro & Rieke, 2010; Lee et al., 2014; Liang & Freed, 2012; Roska & Werblin, 2001). This means that when there is an increase in light (activating the ON pathway), the OFF pathway is suppressed via AC-mediated inhibition. A good example is the AII AC, central to scotopic vision, which provides a crossover inhibition by inhibiting OFF GCs and bipolar cells while simultaneously indirectly exciting ON GCs through their gap junctions with ON bipolar cells (Fig. 2*E*) (Manookin et al., 2008; Masland, 2012).

One of the main functions of crossover inhibition is to produce a sign inversion since it transmits a negative version of the input signal (Jo, Deniz, Xu et al., 2023; Münch et al., 2009; Murphy & Rieke, 2008). This can play a key role in compensating for non-linear rectification at chemical synapses, meaning that cells responding positively to OFF stimulation also respond negatively to ON stimulation (Liang & Freed, 2010; Manookin et al., 2008; Molnar et al., 2009; Rentería et al., 2006; Werblin, 2010). This mechanism ensures that GCs can respond to both types of contrast (light increments and decrements) in a linear, symmetric fashion (Manookin et al., 2008; Molnar et al., 2009).

It has also been reported that crossover inhibition from the ON pathway can contribute to generating the sustained responses observed on the OFF channel. Using two-photon calcium imaging, Rosa et al. (2016) proposed a circuit in zebrafish retinas where GABAergic ACs excited by ON bipolar cells change the temporal properties of the OFF bipolar cells' output, transforming transient, band-pass responses into sustained, low-pass responses. Interestingly, this inhibitory mechanism is further regulated by a disinhibitory network (see the section 'Disinhibition') involving glycinergic ACs.

Traditionally, crossover inhibition has been associated with bistratified ACs, which receive excitatory glutamatergic input from bipolar cells in one layer of the INL and send inhibitory signals to bipolar cells, other ACs and GCs in a different layer, as in the case of AII ACs (Demb & Singer, 2012). However, recently, it has been observed that a new type of AC, the SI-AC, has introduced a novel twist to this process. Unlike bistratified ACs, SI-ACs can process both synaptic inputs and outputs through the same dendrites, enabling localized computation of crossover inhibition within a single dendritic field (Jo, Deniz, Xu et al., 2023).

*Crossover excitation.* Lateral interactions among ACs are not limited to inhibition – some can also be excitatory, either through the release of excitatory neurotransmitters or via gap junctions. Gap junctions enable direct electrical communication between ACs and other retinal neurons, allowing rapid signal transmission (see 'Amacrine cell synapses'). Like crossover inhibition, this process facilitates communication between the ON and OFF pathways by transmitting signals from ON bipolar cells to OFF GCs or vice versa.

Some ACs can even excite specific types of GCs directly through chemical synapses. A key example is ON–OFF VGluT3 ACs, which release glutamate to drive excitatory responses in specific types of GCs (Kim et al., 2015; Lee et al., 2014). These VGluT3 ACs provide ON–OFF crossover excitation to postsynaptic GCs. For instance, when the centre of their receptive field is stimulated, they deliver excitatory input to an ON–OFF and ON direction-selective GC (Lee et al., 2014).

An interesting case occurs when GABAergic transmission is blocked: OFF alpha GCs begin to show an ON response that is dependent on bipolar cells and transmitted via gap junctions (Farajian et al., 2011). This case was named 'masked crossover excitation' in the retina.

*Contrast adaptation.* ACs play a role in contrast adaptation, allowing the visual system to dynamically adjust to varying levels of light contrast in the environment. This adaptation ensures that the retina remains sensitive to changes in contrast (Smirnakis et al., 1997).

Contrast adaptation occurs on multiple timescales, depending on how quickly the adjustment occurs. Fast contrast adaptation, also known as contrast gain control (Victor, 1987), happens over short timescales (Kim & Rieke, 2003), while slow contrast adaptation takes place over longer periods (Smirnakis et al., 1997; Yedutenko et al., 2021). ACs contribute to both mechanisms (Baccus & Meister, 2002; Zaghloul et al., 2005). During slow adaptation, ACs change their polarization in response to high contrast, increasing their activity as contrast levels increase (Baccus & Meister, 2002). In fast adaptation, ACs regulate gain control, normalizing GC responses based on recent visual stimuli (Zaghloul et al., 2005). This mechanism prevents neurons from becoming over-saturated by strong inputs, allowing efficient use of their dynamic range by reducing sensitivity to large, sustained inputs (Laughlin, 1989).

A key element of contrast adaptation is the inhibitory feedback provided by ACs to bipolar cells and GCs. ACs, particularly wide-field ACs, regulate contrast responses by directly inhibiting bipolar cells and modulating synaptic gain (Demb, 2008). This inhibition reduces glutamate release from bipolar cells, effectively adjusting the sensitivity of GCs to contrast fluctuations. Additionally, AC-mediated inhibition can prolong response suppression even after high-contrast stimuli cease, contributing to hyperpolarization-driven after-effects (Demb, 2008).

*Direction selectivity.* ACs are essential for direction selectivity in the retina, a key feature that allows direction-selective GCs to respond preferentially to motion in a specific direction (Demb, 2007; Vaney et al., 2012; Wei, 2018).

Starburst ACs – named for their distinctive radial, starburst-like dendritic arrangement – play a central role in generating direction-selective responses (Barlow & Levick, 1965; Famiglietti, 1991; Tauchi & Masland, 1984; Yoshida et al., 2001). These ACs are themselves direction-selective (Euler et al., 2002) and release GABA onto direction-selective GCs (Fried et al., 2002; Lee et al., 2010; Wei et al., 2011; Yonehara et al., 2011). Their dendrites are aligned in a way that preferentially targets the null direction of motion, suppressing direction-selective GC responses to movement in that direction (Briggman et al., 2011).

When an object moves in a direction-selective GC's 'preferred' direction, starburst ACs provide less inhibition, allowing the GCs to fire more robustly. In contrast, when motion occurs in the 'null' (non-preferred) direction, starburst ACs provide strong GABAergic inhibition, suppressing the direction-selective GCs' firing (Fig. 2*F*) (Demb, 2007; Fried et al., 2002; Vaney et al., 2012; Wei, 2018; Wei et al., 2011). Starburst ACs achieve this directional asymmetry because their dendrites are specialized to detect motion that flows away from the cell body. Each starburst AC's dendrites are more responsive to movement in one direction over others, contributing to the direction selectivity of the entire circuit (Euler et al., 2002). However, other mechanisms can also contribute in parallel to the direction-selectivity. For instance, it has been found that some specific direction-selective GCs display asymmetric dendritic arborizations that orient toward the preferred direction (Trenholm et al., 2011)

*Approaching motion detection.* Some ACs, such as VGlut3 and AII ACs, detect approaching objects (looming detection). Though AII ACs are not inherently selective to approaching motion, they participate in inhibitory circuits that selectively suppress responses to non-approaching objects (Münch et al., 2009). In contrast, VGlut3 ACs are directly tuned to approaching motion and provide glutamatergic excitatory inputs to W3 and tOFFα GCs, two GC types involved in detecting looming threats (Kim et al., 2020) (for circuit mechanism, see Kerschensteiner, 2022).

VGlut3 ACs provide excitatory input to two distinct GC pathways that encode different features of looming objects. W3 GCs respond selectively to looming onset and receive transient excitation from VGlut3 ACs, followed by sustained inhibition from TH2 ACs (Kim & Kerschensteiner, 2017). This inhibitory sequence ensures that W3 GCs maintain consistent response amplitudes, focusing on an object's size rather than its speed. In contrast, tOFFα GCs, which encode looming speed, receive tonic inhibition partially from AII ACs (Kim et al., 2020; Münch et al., 2009). As an object approaches, this inhibition is gradually relieved, allowing tOFFα GCs to track how quickly the stimulus expands. VGlut3 ACs further enhance looming detection through multiple mechanisms. Their excitatory input amplifies RGC responses to stimuli expanding in size, signalling an object moving closer, such as a predator or an approaching obstacle.

Additionally, VGlut3 ACs synchronize excitatory signals to tOFFα RGCs, encoding looming speed, increasing sensitivity to approaching motion and facilitating faster escape responses. Furthermore, VGlut3 ACs shape looming selectivity through dendritic processing, where their proximal dendrites respond strongly to looming stimuli, while their distal dendrites exhibit weaker responses to other types of motion (Kim et al., 2020). Behavioural experiments with mice ablated for VGlut3 ACs showed impaired defensive responses to

looming stimuli, such as flight and prolonged freezing, revealing that VGlut3 ACs are crucial for the innate defensive behaviours to looming visual processing (Kim et al., 2020).

*Object motion detection.* Some types of GCs respond selectively to motion within their receptive field centre, but only if the motion in the surrounding area does not move or follows a different trajectory. In other words, these object motion sensitive (OMS) GCs detect differences in motion patterns between the centre and the surround, allowing the retina to distinguish true object motion from self-induced movements, such as fixational eye movements (jittering motion) (Ölveczky et al., 2003). This ability is crucial for survival, helping with predator avoidance, prey tracking and reacting to approaching threats.

ACs play a key role in this function. Ölveczky et al. (2003) hypothesized that OMS GCs require synchronized peripheral inhibition to trigger an OMS response: excited by motion within or near their receptive field centre but suppressed when the same motion extends into the wider surround. To study it, they also proposed a model for the OMS circuit, where all bipolar cell synapses are rectified, leading to a non-linear spatial integration. The model comprises non-linear subunits in the receptive field centre and a rapid inhibition from a non-linear circuit mediated by ACs in the surround. In a subsequent study (Baccus et al., 2008), the same group further tested the model by recording the responses of various interneurons to global and differential motion. They revealed that polyaxonal ACs exhibit properties suitable for mediating the peripheral inhibition required in OMS. These properties include invariance to the grating phase and rapid, temporally precise depolarization that coincides with the responses of OMS GCs. This suggests that OMS GCs receiving inhibitory input from these ACs will be suppressed whenever motion in the distant periphery matches the motion within the receptive field centre.

Advances in genetic tools for targeting specific AC types have expanded our understanding of their role in OMS and their underlying circuit (see more in Kerschensteiner, 2022). For instance, VGlut3 ACs increase activity when an object moves independently of its background but decrease activity when the object and background move together. These ACs will then inhibit W3 GCs, conferring on them object-motion sensitivity (Kim et al., 2015). VGlut3 ACs – despite having larger dendritic fields than bipolar cells – can process inputs locally, ensuring high spatial precision in object motion detection (Chen et al., 2017; Hsiang et al., 2017). VGlut3 ACs may also introduce a delayed excitatory signal during motion, enabling surround inhibition to suppress centre excitation during global image motion (Krishnaswamy et al., 2015), or

enhancing OMS responses by adding surround inhibition to the excitatory pathway (Kim & Kerschensteiner, 2017; Kim et al., 2015).

In contrast, another type of AC, the TH2 AC, controls W3 RGCs responses by selectively inhibiting global motion while preserving responses to local motion (Kim & Kerschensteiner, 2017). These cells respond to both motion types but distinguish them through response kinetics: global motion rapidly activates TH2 ACs, triggering quick GABA release that suppresses W3 RGCs, whereas local motion gradually depolarized TH2 ACs, thereby preventing GABA release and allowing W3 RGC activity (Kim & Kerschensteiner, 2017).

*Anticipation.* When the retina is stimulated by a moving bar at a constant speed, the population activity of GCs does not lag behind the bar's movement. Instead, their activity remains aligned with the bar's position, as though the retina anticipates its motion. This phenomenon is significant because GCs typically respond with a latency of about 50 ms due to delays from phototransduction and synaptic transmission.

Berry et al. (1999) first described this feature, showing that when a dark moving bar stimulates the retina, the activity of fast OFF ganglion cells aligns with the leading edge of the bar rather than lagging as expected from response delays. This finding demonstrated the existence of a mechanism that compensates for the inherent delays and accurately represents the bar's position. This effect was initially observed in salamander and rabbit retinas using multi-electrode array recordings and has since been identified in goldfish (Johnston & Lagnado, 2015) and mouse retinas (DePiero & Borghuis, 2022)

Although the exact cell type responsible for motion anticipation remains unknown, ACs are strong candidates. Electrophysiological recordings and modelling in goldfish retinas demonstrated that motion anticipation relies on feedforward inhibition from ACs (Johnston & Lagnado, 2015). This inhibition interacts non-linearly with excitatory inputs in GC dendrites, shifting their responses forward in time to encode the actual position of a moving bar rather than its delayed representation. Inhibitory input is essential for suppressing trailing excitation, ensuring that the stimulus is accurately encoded as it moves across the receptive field.

*Omitted stimulus response.* When a sequence of flashes is presented, some GCs respond specifically to the omission of the next flash. Interestingly, the latency of this response adjusts with the frequency of the flash sequence, suggesting that the GCs maintain a constant latency to the expected – but missing – stimulus. For this reason, this has been named the omitted stimulus response (OSR) (Schwartz et al., 2007). The OSR demonstrates the retina's ability to predict the timing of future stimuli based on pre-

vious patterns. This change in latency of the predictive response disappears when glycinergic transmission is blocked, indicating that glycinergic ACs are critical for this function. Modelling data suggest that the synaptic depression of glycinergic ACs plays a crucial role in changing the latency of the OSR when the frequency of the flash sequence is changed (Ebert et al., 2024). Even though the specific cells involved in this microcircuit remain unidentified, this discovery shows that the depressing nature of the inhibitory synapses can mediate complex computations performed by the retina.

**Conclusion.** ACs play a critical and complex role in pre-processing visual information to GCs before it is sent to the brain. While several well-studied AC types, such as starburst and AII ACs, have specialized roles in distinct retinal circuits, the functional role of many other AC types remains unclear (see Table 2). As discussed in previous sections of this review, several connectivity motifs have been identified that link structure to function. However, the specific AC subtypes involved in many of these motifs are largely unknown, with only a few exceptions. Despite the identification of more than 60 AC types, only a tiny fraction have been extensively studied, and therefore it remains uncertain whether each type serves a distinct, isolated function or whether they interact beyond their immediate microcircuits (Franke & Baden, 2017)

### General conclusion

While our knowledge of the retina surpasses that of many other regions of the mammalian central nervous system, significant gaps remain, particularly regarding ACs. Compared to photoreceptors and bipolar cells, which are relatively well-studied, ACs remain less understood despite their critical role in retinal function. Although most studies on ACs have focused on well-known types such as AII, starburst and A17, they represent only a fraction of the total AC population. Recent works have identified additional types with unique morphological and physiological characteristics. However, their specific roles in retinal circuits remain largely unknown (Akrouh & Kerschensteiner, 2015; Park et al., 2015; Zhu et al., 2014). In Table 1 and Table 2, we have compiled information on nearly 30 AC types out of the 67 identified. For many of these, even basic features such as their pre- and post-synaptic targets are still unknown. Understanding these and other underexplored AC types will require advances in transcriptomic, anatomical and functional studies. The development of techniques such as calcium and voltage imaging, specific promoters and genetic mouse lines has substantially improved our understanding of ACs, and we expect that even more progress can be made by combining these new tools in the future.

ACs represent one of the most diverse and functionally significant neuronal populations in the retina, with remarkable morphology, physiology and connectivity diversity. Recent research has revealed intriguing insights into their possible organization and function, as suggested by recent calcium imaging experiments on GABAergic ACs that showed that their dendrites have up to 25 types of chromatic responses (Korympidou et al., 2024). While we aimed to provide a broad perspective on the diversity of AC morphology, connectivity and function, fully capturing all aspects of this complex cell class is challenging. One particularly important feature we do not discuss in this review is synaptic plasticity involving ACs. Many reviews have explored various forms of plasticity in the retina, and we encourage readers to refer to them for a deeper understanding into AC involvement on plasticity: neuromodulation (McMahon & Dowling, 2023), electrical synapses (O'Brien & Bloomfield, 2018) and short-term plasticity (Deng et al., 2024).

A broader question is how recurrent the amacrine cell network is. Beyond the classical micro-circuits listed above, several studies, particularly those highlighting serial inhibition (disinhibitory loops), suggest that the network of ACs is so interconnected. It is unclear if the functions of this network can be entirely described with a micro-circuit approach described above, where two or three cell types interact to form a specific computation. Single cell stimulation of bipolar cells (Asari & Meister, 2012, 2014; Spampinato et al., 2022) showed that many GC types could be reached from stimulation of a single bipolar cell, possibly going through ACs. If a single cell stimulation can modulate most cell types, this advocates for a different view. We might need to consider the AC network as a full-blown recurrent network, bearing some similarities with the balanced networks that have been used to study the cortex, with excitatory and inhibitory neurons being recurrently connected (Ahmadian & Miller, 2021; Kadmon et al., 2020; Miller & Palmigiano, 2020; Podlaski & Machens, 2024; van Vreeswijk & Sompolinsky, 1996). In a first approximation, ACs might be similar to a recurrent network of inhibitory cells, with no recurrent excitatory connections like the classical balanced networks. However, as discussed above, each cell might have many subunits. These subunits may be partially coupled to each other because they are part of the same cell or connected through gap junctions. These couplings might play a role similar to recurrent excitation. Single-cell stimulation of well-defined AC types will shed light on this issue. This perspective might change our view on the retinal circuit. Instead of a mostly feed-forward network, it might be better seen as a recurrent network that is read out by GCs.

Finally, emerging genetic tools and techniques, such as optogenetics and two-photon holographic stimulation, provide exciting opportunities to selectively manipulate

specific AC types and explore their roles in retinal processing (Beier et al., 2013; Spampinato et al., 2022). Combining these approaches with connectomic and computational modelling will be essential for unravelling AC functions in the retina. These advancements promise to deepen our understanding of how the retina encodes and processes the visual world, shifting the simplistic view of ACs from simple lateral inhibitory elements to dynamic, integral components of a sophisticated neural network.

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

## Additional information

### Competing interests

The authors declare that they have no conflicts of interest with the contents of this article.

### Author contributions

V.C.G: conception and design; collection and assembly of data; data analysis and interpretation; manuscript writing. D.V: conception and design; collection and assembly of data; data analysis and interpretation; manuscript writing. T.B: data analysis and interpretation; manuscript writing. O.M: conception and design; administrative support; financial support, data analysis and interpretation; manuscript writing. All authors have read and approved the final version of this manuscript and agree to be accountable for all aspects of the work in ensuring that questions related to the accuracy or integrity of any part of the work are appropriately investigated and resolved. All persons designated as authors qualify for authorship, and all those who qualify for authorship are listed.

### Funding

This work has been funded by an ERC grant (No. 101045253, DEEPRETINA) to O.M. T.B. was funded by a PhD fellowship from ENS and supported by the Fondation pour la Recherche Médicale, grant number FDT202304016465.

### Acknowledgements

The authors would like to thank the entire team for their valuable discussions, which were crucial in shaping the ideas presented in this manuscript.

### Keywords

physiology, retina, retinal physiology, vision

## Supporting information

Additional supporting information can be found online in the Supporting Information section at the end of the HTML view of the article. Supporting information files available:

**Peer Review History**

