## [Peer Review History · The Journal of Physiology]

The Mysterious Middlemen Making Your Vision Pop: Understanding the Function of Amacrine Cells

Victor Calbiague, Deborah Varro, Thomas Buffet, and Olivier Marre
DOI: 10.1113/JP287958

Corresponding author(s): Victor Calbiague (victor-manuel.calbiague-garcia@inserm.fr)

Review Timeline:

Submission Date:	29-Dec-2024
Editorial Decision:	28-Jan-2025
Revision Received:	05-Mar-2025
Accepted:	31-Mar-2025

Senior Editor: Laura Bennet

Reviewing Editor: Karin Dedek

Transaction Report:

Dear Dr Calbiague,

Re: JP-TR-2024-287958 **"The Mysterious Middlemen Making Your Vision Pop: Comprehensive Overview to Understanding The Function of Amacrine Cells"** by Victor Calbiague, Deborah Varro, Thomas Buffet, and Olivier Marre

Thank you for submitting your manuscript to The Journal of Physiology. It has been assessed by a Reviewing Editor and by 2 expert referees and we are pleased to tell you that it is acceptable for publication following satisfactory revision.

ABSTRACT FIGURES: Authors may use The Journal's premium BioRender account to create/redraw their Abstract Figures (and any other suitable schematic figure). Information on how to access this account is here: <https://physoc.onlinelibrary.wiley.com/journal/14697793/biorender-access>.

REVISION CHECKLIST: Upload a full Response to Referees file. To create your 'Response to Referees' copy all the reports, including any comments from the Senior and Reviewing Editors, into a Microsoft Word, or similar, file and respond to each point, using font or background colour to distinguish comments and responses and upload as the required file type.

We look forward to receiving your revised submission.

Yours sincerely,

Laura Bennet
Senior Editor

EDITOR COMMENTS

Reviewing Editor:

Dear authors,

Your manuscript has been reviewed by two reviewers and myself. We all agree that the existing literature on amacrine cells, their types and function is well summarized and will be present a useful resource for the field. As you will see from the reviewers' comments, some terms and statements need to be refined.

Please also see 'Required Items' below.

Best wishes

Your Reviewing Editor

REFEREE COMMENTS

Referee #1:

This review provides an extensive overview of the literature documenting the role and diversity of amacrine cells in the retina. The focus is largely on mammalian/mouse retina, which is appropriate given the importance of using genetic data to define cell-types and genetic manipulations to probe the functional roles of neurons.

The commentary documents the main roles and circuit motifs and points to general areas for future investigation. Table 1 provides a nice overview, with plenty of empty boxes to be filled by future studies. The narrative structure works well. A general comment is that more detail could have been provided regarding the circuit mechanisms underlying the different circuit motifs. Just as an example, the anticipatory response (4.2.11) is described, but the circuit mechanisms are not. On the other hand, the review is already lengthy, so maybe it is appropriate to describe the motifs while providing the reader with the key references to dig into the details if they want to.

Other comments

L95: "*Before genetic tools, the main classification of AC subtypes typically was based on dendritic morphology.* This statement discounts early use of immunohistochemical markers, to identify AC types, (cholinergic, serotonergic, VIP, NOS, Subst P, DACs, etc), which amounts to using molecular/genetic rather than anatomical characteristics.

L225: "*...the expression profile of ion channels reflects the electrical properties and firing patterns of each AC type...*". Doesn't the latter reflect the former?

L663-669 : "*... This phenomenon, called "oversaturation," occurs when the retinal cells, especially those sensitive to low-light stimuli, e.g., rod photoreceptors, are saturated by high-intensity light. To prevent this, ACs may reduce gap junction coupling, which diminishes the spread of electrical signals between cells (Feigenspan et al., 2001). By limiting this coupling, ACs help the retina maintain sensitivity to a wide range of light intensities, ensuring proper visual function and adaptation to changing lighting conditions.*"

The distinction between "saturation" and "oversaturation" should be explained in more detail. Modulating the strength of coupling controls the extent of signal spread - but how is this tied to the amplitude/sensitivity of responses and how does oversaturation come into play?

Paragraph starting L670: It would be helpful to also discuss the trade-offs inherent to coupling, e.g. the impact on spatial resolution.

L685: Does this dual effect of inhibition and gap-junction excitation only apply to All ACs or do other AC types also show such functional diversity? If so, it would be helpful to document all such types, if not, then make it clear that this only applies to All ACs.

L787: Disinhibition provides a major excitatory drive to Off-alpha GCs. The authors might consider including a discussion of this motif as an example of direct disinhibition.

Referee #2:

This manuscript article by Calbiague-Garcia and colleagues provides a comprehensive review of amacrine cell (AC) types and functions in the vertebrate retina. This is not an easy task, given the complexity and diversity of amacrine cell types. The authors should be applauded by devoting a substantial amount of work to summarize this broad topic. They did a nice job in trying to provide an almost exhaustive description of amacrine cells, which will be a valuable resource for researchers who are interested in amacrine cells and retinal circuitry. I have some comments and suggestions below:

1. While this manuscript covers a broad range of AC structure and function, understandably, it is impossible to include everything. In this regard, I recommend mentioning to readers some recent review articles that discuss amacrine cells from different perspectives to fill in some gaps. A couple of examples are reviews on neuromodulatory signaling by amacrine cells by McMahon and Dowling, 2023 (PMCID: PMC10625386) and synaptic plasticity and amacrine cell function by Deng et al., 2024 (PMCID: PMC11414732).

2. Interpretation of molecular cell types:

Line 254: "A comparable number of AC clusters were identified in both the foveal (27 clusters) and peripheral regions (34 clusters) of the macaque retina (Peng et al., 2019), reinforcing the idea that AC diversity is conserved across species (Hahn et al., 2023).".

This statement is not supported by the numbers: there are about 60 AC clusters in the mouse (Yan et al., 2020, PMCID: PMC7329304), compared to ~ 30 clusters in the macaque, I wouldn't call that "a comparable number".

Line 259: "Its relevance is underscored by recent findings that retinal molecular architecture is highly conserved across 17 vertebrate species (Hahn et al., 2023), suggesting that specific molecular markers can be used across species to target analogous cell types."

I think here the authors confused "analogous cell types" with "homologous cell types". Homologous structures share common evolutionary origin but can have very different functions (e.g. wings vs hands), while analogous structures share similar functions but can have very different evolutionary origins (e.g. wings of birds vs wings of flies). The Hahn et al., 2023 paper demonstrates the conservation of homologous (or orthologous) cell classes (not cell types) in the context of single-cell RNA seq data, it is not about the conservation of analogous cell types. In fact, the number of RGC types and AC types differ dramatically between primates and mice, arguing against cell-type specific markers that are conserved across species (discussed in detail in Shekhar and Sanes, 2021). This poor conservation of RGC molecular markers across species was

directly demonstrated by Peng et al., 2019 Fig. 5 (PMCID: PMC6424338) and by Dhande et al., 2019 on a specific RGC type (PMCID: PMC6325260).

3. Line 317: CHAT is a protein, not a neuropeptide.

4. Line 436: "... allowing the information to either bypass or not the soma" Grammar mistake.

5. Lines 441-444: "In contrast, each dendrite of starburst ACs can independently detect motion in a specific direction due to the localized excitatory and inhibitory synaptic inputs it receives (Euler et al., 2002).

Euler's 2002 paper is the first study to demonstrate that individual dendrites of starburst cells are tuned to different motion directions, but it didn't demonstrate that it's "due to localized excitatory and inhibitory inputs". In fact, localized excitatory and inhibitory inputs are not the main mechanism of starburst direction selectivity.

"The direction-selective response emerges from computation within dendrites and the interplay between dendritic excitatory inputs (Wei & Feller, 2011)."

Again, this statement and the reference need revision. Multiple synaptic and dendritic mechanisms underlie the direction selectivity of starburst amacrine cells, as reported by tens of papers. A fairly recent review covering this topic is Wei, *Annu Rev Vis Sci.* 2018.

6. Line 479-480: I suggest citing the most recent paper on voltage imaging in the starburst amacrine cell dendrites here: Acarón Ledesma et al., 2024.

7. Line 487: Suggest removing the Wei and Feller 2011 reference.

8. Line 532: "For instance, bistratified narrow-field starburst ACs which are crucial..": This is incorrect, starburst ACs are monostratified medium field cells.

9. Line 761: Disinhibition: I suggest including the Chen et al., 2020 paper (PMCID: PMC7728437), which shows that serial inhibition can implement computations other than disinhibition.

10. Line 914: The more definitive references on direction-selective GABA release from starburst to direction-selective ganglion cells are: Lee et al., 2010; Wei et al., 2011; Yonehara et al., 2011. These references should be added together with Fried et al., 2002.

11. Line 920: The reference Schachter et al., 2010 (which is on dendritic spike generation) seems to be randomly inserted here. For the statement that "Starburst ACs provide strong GABAergic inhibition, suppressing the direction-selective GCs's firing", there are dozens of references that support this statement from decades of research. I suggest to include a review article here, or add landmark papers in the DS field.

12. Line 973: Section 4.2.11. Anticipation: It is unclear if ACs are relevant for this computation.

REQUIRED ITEMS

- Please include an Abstract Figure file, as well as the Figure Legend text within the main article file. The Abstract Figure is a piece of artwork designed to give readers an immediate understanding of the Review Article and should summarise the main conclusions. If possible, the image should be easily 'readable' from left to right or top to bottom. It should show the physiological relevance of the Review so readers can assess the importance and content of the article. Abstract Figures should not merely recapitulate other figures in the Review. Please try to keep the diagram as simple as possible and without superfluous information that may distract from the main conclusion of the Review. Abstract Figures must be provided by authors no later than the revised manuscript stage and should be uploaded as a separate file during online submission labelled as File Type 'Abstract Figure'. Please ensure that you include the figure legend in the main article file. All Abstract Figures will be sent to a professional illustrator for redrawing and you may be asked to approve the redrawn figure before your paper is accepted.

- Your MS must include a complete "Additional information section" with the following 4 headings and content:

Competing Interests: A statement regarding competing interests. If there are no competing interests, a statement to this effect must be included. All authors should disclose any conflict of interest in accordance with journal policy.

Author contributions: Each author should take responsibility for a particular section of the study and have contributed to writing the paper. Acquisition of funding, administrative support or the collection of data alone does not justify authorship; these contributions to the study should be listed in the Acknowledgements. Additional information such as 'X and Y have contributed equally to this work' may be added as a footnote on the title page.

It must be stated that all authors approved the final version of the manuscript and that all persons designated as authors qualify for authorship, and all those who qualify for authorship are listed.

Funding: Authors must indicate all sources of funding, including grant numbers. If authors have not received funding, this must be stated.

It is the responsibility of authors funded by RCUK to adhere to their policy regarding funding sources and underlying research material. The policy requires funding information to be included within the acknowledgement section of a paper. Guidance on how to acknowledge funding information is provided by the Research Information Network. The policy also requires all research papers, if applicable, to include a statement on how any underlying research materials, such as data, samples or models, can be accessed. However, the policy does not require that the data must be made open. If there are considered to be good or compelling reasons to protect access to the data, for example commercial confidentiality or legitimate sensitivities around data derived from potentially identifiable human participants, these should be included in the statement.

Acknowledgements: Acknowledgements should be the minimum consistent with courtesy. The wording of acknowledgements of scientific assistance or advice must have been seen and approved by the persons concerned. This section should not include details of funding.

- Please upload separate high quality figure files via the submission form.

- Author profile(s) must be uploaded via the submission form. Authors should submit a short biography (no more than 100 words for one author or 150 words in total for two authors) and a portrait photograph of the two leading authors on the paper. These should be uploaded and clearly labelled together in a Word document with the revised version of the manuscript. Any standard image format for the photograph is acceptable, but the resolution should be at least 300 DPI and preferably more. A group photograph of all authors is also acceptable, providing the biography for the whole group does not exceed 150 words.

- Please ensure that the Article File you upload is a Word file.

- The corresponding author must provide an institutional email address (not a personal address) for their author account. We encourage ALL co-authors to also provide institutional email addresses. If this cannot be provided (as corresponding author), then a stamped letter must be provided from the institution which confirms their role and employment there (please upload this with the revised submission).

END OF COMMENTS

Dear Editor,

We sincerely appreciate your thorough review of our manuscript, as well as your positive comments and valuable feedback. In our efforts to enhance the quality of our work, we have carefully addressed all your concerns and incorporated additional corrections.

We hope these revisions have significantly improved our manuscript and made it suitable for publication.

Best regards,
Victor Calbiague-Garcia

Response to Referees

Reviewing Editor:

Dear authors,

Your manuscript has been reviewed by two reviewers and myself. We all agree that the existing literature on amacrine cells, their types and function is well summarized and will be present a useful resource for the field. As you will see from the reviewers' comments, some terms and statements need to be refined.

Please also see 'Required Items' below.

Best wishes

Your Reviewing Editor

REFEREE COMMENTS

Referee #1:

This review provides an extensive overview of the literature documenting the role and diversity of amacrine cells in the retina. The focus is largely on mammalian/mouse retina, which is appropriate given the importance of using genetic data to define cell-types and genetic manipulations to probe the functional roles of neurons.

1) The commentary documents the main roles and circuit motifs and points to general areas for future investigation. Table 1 provides a nice overview, with plenty of empty boxes to be filled by future studies. The narrative structure works well. A general comment is that more detail could have been provided regarding the circuit mechanisms underlying the different circuit motifs. Just as an example, the anticipatory response (4.2.11) is described, but the circuit mechanisms are not. On the other hand, the review is already lengthy, so maybe it is appropriate to describe the motifs while providing the reader with the key references to dig into the details if they want to.

R = We appreciate the insightful comment. We agree that our manuscript lacked details on circuit mechanisms in the last section. To address this, we have added a brief explanatory paragraph in each relevant section to outline the circuit motifs underlying contrast adaptation

(line 909), direction selectivity (line 917), approaching motion (line 938), object motion detection (line 988), motion anticipation (line 1019), and response to omitted stimuli (line 1027). Additionally, we have included references to guide readers who wish to explore these motifs in greater depth.

Other comments

2) L95: "Before genetic tools, the main classification of AC subtypes typically was based on dendritic morphology". This statement discounts early use of immunohistochemical markers, to identify AC types, (cholinergic, serotonergic, VIP, NOS, Subst P, DACs, etc), which amounts to using molecular/genetic rather than anatomical characteristics

R = To ensure a more comprehensive historical perspective, we have revised the paragraph to highlight the contribution of these markers (e.g., cholinergic, serotonergic, VIP, NOS, Substance P, DACs) alongside morphological criteria in amacrine cell classification.

Line 103: *'Before the development of genetic tools, AC classification relied primarily on dendritic morphology. However, it is essential to mention that early immunohistochemical techniques further improve this classification by using molecular markers—such as cholinergic, serotonergic, VIP, NOS, Substance P, and dopamine—to distinguish AC subtypes beyond their anatomical features'*

3) L663-669 : "... This phenomenon, called "oversaturation," occurs when the retinal cells, especially those sensitive to low-light stimuli, e.g., rod photoreceptors, are saturated by high-intensity light. To prevent this, ACs may reduce gap junction coupling, which diminishes the spread of electrical signals between cells (Feigenspan et al., 2001). By limiting this coupling, ACs help the retina maintain sensitivity to a wide range of light intensities, ensuring proper visual function and adaptation to changing lighting conditions."

The distinction between "saturation" and "oversaturation" should be explained in more detail. Modulating the strength of coupling controls the extent of signal spread - but how is this tied to the amplitude/sensitivity of responses and how does oversaturation come into play?

R = We acknowledge that the distinction between "saturation" and "oversaturation" was not clearly defined, and we realize that our interpretation of that paragraph may have been inaccurate. To clarify this, we have revised the paragraph and decided to modify that section by removing the specific part in question.

Line 646: *'ACs form gap junctions that connect with neighboring cells, such as other ACs, bipolar cells, and GCs (Pan et al., 2010). These gap junctions are essential for regulating retinal responses to changes in light intensity. The strength of gap junction coupling between ACs can be dynamically modulated in response to environmental stimuli, such as varying light levels (Curti & O'Brien, 2016). Depending on ambient light conditions, ACs adjust their coupling to regulate retinal responses to different light intensities and contrasts. This regulation helps the retina maintain sensitivity across a wide range of light levels, ensuring efficient visual function and adaptation to changing lighting conditions'*

4) Paragraph starting L670: It would be helpful to also discuss the trade-offs inherent to coupling, e.g. the impact on spatial resolution.

R = We have incorporated a discussion on the consequences of electrical coupling, particularly its impact on spatial resolution. Below is the revised paragraph, which now

highlights how coupling influences the balance between sensitivity and visual detail across different lighting conditions.

Line 654: *'Gap junctions also enable rapid synchronization of electrical activity across large networks of cells. All ACs, which are part of the rod pathway, are activated by rod bipolar cells and are critical for transmitting rod-driven information during scotopic vision. These cells are electrically coupled to one another and to cone bipolar cells, forming a network that efficiently relays rod-driven signals to GCs (Bloomfield & Völgyi, 2009; Demb & Singer, 2012). However, a key disadvantage of electrical coupling is its impact on spatial resolution. Because All ACs are strongly coupled to one another and to cone bipolar cells, electrical signals from individual photoreceptors can spread laterally across the network. While this lateral spread enhances sensitivity under low-light conditions by integrating dim light signals, it also causes spatial blurring as fine visual details are averaged out. This trade-off is particularly significant in scotopic conditions, where electrical coupling is higher than in photopic conditions (Vardi & Smith, 1996; Dunn et al., 2006). In low-light situations, maximizing sensitivity and reducing noise take precedence, even if spatial detail is compromised. In contrast, during bright-light conditions, reduced electrical coupling enhances spatial resolution, preserving fine visual details. In conclusion, the retina adjusts gap junction coupling based on ambient light levels to strike a balance between sensitivity and resolution. Interestingly, the strongest coupling is observed under intermediate light conditions, optimizing the signal-to-noise ratio and facilitating luminance adaptation (Bloomfield & Völgyi, 2009). This plasticity is also influenced by factors like dopamine levels and circadian rhythms, which regulate connexin phosphorylation and dephosphorylation (Bloomfield & Völgyi, 2009).'*

5) L685: Does this dual effect of inhibition and gap-junction excitation only apply to All ACs or do other AC types also show such functional diversity? If so, it would be helpful to document all such types, if not, then make it clear that this only applies to All ACs.

R = To provide a more comprehensive overview, we have documented all amacrine cell types for which we found evidence of both inhibitory and gap-junction-mediated excitatory effects, along with the corresponding references.

Line 683: *'The combined inhibitory and excitatory signaling through gap junctions has been described in various AC types, including the All AC (Hartveit and Veruki 2012), the A17 ACs (Elgueta et al. 2018), the A8 ACs (Lee et al. 2015; Yadav et al. 2019), the nNOS2 ACs (Jacoby et al. 2018; Zhu et al. 2014), and the VIP ACs (Park et al. 2015).'*

6) L787: Disinhibition provides a major excitatory drive to Off-alpha GCs. The authors might consider including a discussion of this motif as an example of direct disinhibition.

R = We appreciate the referee's suggestion and have incorporated a discussion of this motif as an example of direct disinhibition in paragraph XXXX:

Line 760: Disinhibition also plays a major role in driving excitatory responses in OFF GCs by regulating their sensitivity to light decrements (Manookin et al., 2008; Margulis and Detwiler 2007; Van Wyk et al., 2009). When light decreases, ON bipolar cells hyperpolarize, leading to a reduction in excitatory input to All ACs, which in turn reduces their glycine release onto OFF GCs. This reduction in inhibition, or disinhibition, results in an inward current that enhances the response of OFF GCs. This effect is particularly strong at low contrast, contributing significantly to the overall response. However, as contrast increases, the role of disinhibition diminishes relative to direct excitation from OFF bipolar cells. This mechanism is

essential for shaping the sensitivity of OFF GCs to visual stimuli, especially at low contrast, and ensures effective processing of contrast changes in the visual system.'

Referee #2:

This manuscript article by Calbiague-Garcia and colleagues provides a comprehensive review of amacrine cell (AC) types and functions in the vertebrate retina. This is not an easy task, given the complexity and diversity of amacrine cell types. The authors should be applauded by devoting a substantial amount of work to summarize this broad topic. They did a nice job in trying to provide an almost exhaustive description of amacrine cells, which will be a valuable resource for researchers who are interested in amacrine cells and retinal circuitry. I have some comments and suggestions below:

7) While this manuscript covers a broad range of AC structure and function, understandably, it is impossible to include everything. In this regard, I recommend mentioning to readers some recent review articles that discuss amacrine cells from different perspectives to fill in some gaps. A couple of examples are reviews on neuromodulatory signaling by amacrine cells by McMahon and Dowling, 2023 (PMCID: PMC10625386) and synaptic plasticity and amacrine cell function by Deng et al., 2024 (PMCID: PMC11414732).

R = We appreciate the referee's suggestion. To provide readers with additional perspectives on amacrine cell function, we have added a paragraph in the general conclusion section citing the recommended review articles by McMahon and Dowling (2023) and Deng et al. (2024). Additionally, we have included the review by O'Brien and Bloomfield (2018) on plasticity in connexins.

Line 1072: *'...While we aimed to provide a broad perspective on the diversity of AC morphology, connectivity, and function, fully capturing all aspects of this complex cell class is challenging. One particularly important feature we do not discuss in this review is synaptic plasticity involving ACs. Many reviews have explored various forms of plasticity in the retina, and we encourage readers to refer to them for a deeper understanding into AC involvement on plasticity: neuromodulation (McMahon and Dowling, 2023), electrical synapses (O'Brien and Bloomfield, 2018), and short-term plasticity (Deng et al., 2024).'*

8) Interpretation of molecular cell types:

Line 254: "A comparable number of AC clusters were identified in both the foveal (27 clusters) and peripheral regions (34 clusters) of the macaque retina (Peng et al., 2019), reinforcing the idea that AC diversity is conserved across species (Hahn et al., 2023)."

This statement is not supported by the numbers: there are about 60 AC clusters in the mouse (Yan et al., 2020, PMCID: PMC7329304), compared to ~ 30 clusters in the macaque, I wouldn't call that "a comparable number".

R = We acknowledge that the numbers are evidently not similar, with nearly twice as many AC clusters reported in mice compared to primates. However, similar to the case in mice, there is a possibility that the number of amacrine cell types in primates has been underestimated due to sample size limitations. To address this, we have revised the text to highlight this potential source of discrepancy. Additionally, we have incorporated the hypothesis proposed by Shekhar and Sanes (2018), which suggests that the difference may stem from primates relying more on cortical processing than retinal computations:

Line 258: *'Interestingly, fewer AC clusters have been identified in primates, with 27 clusters in the fovea and 34 in the periphery of the macaque retina (Peng et al., 2019), as opposed to approximately 60 in the mouse retina. This disparity could be due to sample size limitations, as early studies of mouse retinas showed that analyzing fewer cells often resulted in fewer identified subtypes, suggesting that AC diversity in primates may have been similarly underestimated. Alternatively, a teleological explanation suggests that species such as mice, chicks, and fish rely more on retinal processing, while primates depend more on cortical processing (Shekhar & Sanes, 2021).'*

9) Line 259: "Its relevance is underscored by recent findings that retinal molecular architecture is highly conserved across 17 vertebrate species (Hahn et al., 2023), suggesting that specific molecular markers can be used across species to target analogous cell types."

I think here the authors confused "analogous cell types" with "homologous cell types". Homologous structures share common evolutionary origin but can have very different functions (e.g. wings vs hands), while analogous structures share similar functions but can have very different evolutionary origins (e.g. wings of birds vs wings of flies). The Hahn et al., 2023 paper demonstrates the conservation of homologous (or orthologous) cell classes (not cell types) in the context of single-cell RNA seq data, it is not about the conservation of analogous cell types. In fact, the number of RGC types and AC types differ dramatically between primates and mice, arguing against cell-type specific markers that are conserved across species (discussed in detail in Shekhar and Sanes, 2021). This poor conservation of RGC molecular markers across species was directly demonstrated by Peng et al., 2019 Fig. 5 (PMCID: PMC6424338) and by Dhande et al., 2019 on a specific RGC type (PMCID: PMC6325260).

R = We fully agree with the referee and acknowledge the misuse of the terms. To ensure clarity, we have revised the paragraph to explicitly state "*homologous cell subclasses*" instead, avoiding any potential confusion.

Line 266: *'The transcriptome-based classification mentioned above provides a comprehensive atlas for targeting AC types and establishes a standardized reference across laboratories. Its relevance is underscored by recent findings that retinal molecular architecture is highly conserved across 17 vertebrate species (Hahn et al., 2023), which could be that specific molecular markers can be used across species to target homologous cell subclasses.'*

10) Line 317: CHAT is a protein, not a neuropeptide.

11) Line 436: "... allowing the information to either bypass or not the soma" Grammar mistake.

R = We have addressed the errors identified by the referee.

12) Lines 441-444: "In contrast, each dendrite of starburst ACs can independently detect motion in a specific direction due to the localized excitatory and inhibitory synaptic inputs it receives (Euler et al., 2002).

Euler's 2002 paper is the first study to demonstrate that individual dendrites of starburst cells are tuned to different motion directions, but it didn't demonstrate that it's "due to localized excitatory and inhibitory inputs". In fact, localized excitatory and inhibitory inputs are not the main mechanism of starburst direction selectivity.

13) "The direction-selective response emerges from computation within dendrites and the interplay between dendritic excitatory inputs (Wei & Feller, 2011)."

Again, this statement and the reference need revision. Multiple synaptic and dendritic mechanisms underlie the direction selectivity of starburst amacrine cells, as reported by tens of papers. A fairly recent review covering this topic is Wei, *Annu Rev Vis Sci.* 2018.

R (For points 12 and 13) = We have revised the paragraph to ensure accuracy in describing the mechanisms underlying starburst amacrine cell direction selectivity:

Line 436: *'In contrast, each dendrite of starburst ACs independently detects motion in a specific direction (Euler et al., 2002). This directional selectivity arises from a combination of dendritic morphology, synaptic inputs, and the passive and active membrane properties that shape the centrifugal preference of starburst AC dendrites (for an in-depth discussion of different models of centrifugal direction selectivity in starburst ACs, see Wei, 2018).'*

14) Line 479-480: I suggest citing the most recent paper on voltage imaging in the starburst amacrine cell dendrites here: Acarón Ledesma et al., 2024.

15) Line 487: Suggest removing the Wei and Feller 2011 reference

16) Line 532: "For instance, bistratified narrow-field starburst ACs which are crucial..": This is incorrect, starburst ACs are monostratified medium field cells.

R = We acknowledge the referee's suggestions and have implemented the requested changes. We have included the reference to Acarón Ledesma et al. (2024), removed the Wei and Feller (2011) citation, and corrected the description of starburst amacrine cells to accurately indicate that they are monostratified medium-field cells.

17) . Line 761: Disinhibition: I suggest including the Chen et al., 2020 paper (PMCID: PMC7728437), which shows that serial inhibition can implement computations other than disinhibition.

R = We have added the reference to Chen et al. (2020) to acknowledge that serial inhibition can support computations beyond disinhibition:

Line 778: *'As we have discussed, disinhibitory motifs typically involve two inhibitory neurons in sequence, where inhibiting the second neuron results in an excitatory effect on the principal neuron. However, Chen et al. (2020) revealed an additional function of a disinhibitory microcircuit in the retina, challenging the conventional view that disinhibition simply reduces inhibition. Instead, their study demonstrated that serial inhibition within the starburst AC-direction selective GC microcircuit does not relieve inhibition but instead preserves it through an interaction between network dynamics and short-term synaptic plasticity. Their study showed that, under noisy visual conditions, starburst ACs receive inhibitory input from neighboring starburst ACs. This suppression is essential because it prevents excessive activation of starburst ACs before a motion stimulus, thereby avoiding short-term synaptic depression at starburst AC – Direction selective GC synapses. Without this regulation, starburst AC-mediated inhibition of direction-selective GC becomes weaker due to synaptic depression, ultimately impairing direction selectivity.'*

18) Line 914: The more definitive references on direction-selective GABA release from starburst to direction-selective ganglion cells are: Lee et al., 2010; Wei et al., 2011; Yonehara et al., 2011. These references should be added together with Fried et al., 2002.

19) Line 920: The reference Schachter et al., 2010 (which is on dendritic spike generation) seems to be randomly inserted here. For the statement that "Starburst ACs provide strong GABAergic inhibition, suppressing the direction-selective GCs's firing", there are dozens of references that support this statement from decades of research. I suggest to include a review article here, or add landmark papers in the DS field.

R = We have incorporated the recommended references to strengthen the statement in line 914. Additionally, we have replaced the Schachter et al. (2010) reference, as noted by the referee, and have included other relevant sources to support the claim: Fried et al. (2002), Wei et al. (2010), Vaney et al. (2012), Wei (2018), and Demb (2007).

20) Line 973: Section 4.2.11. Anticipation: It is unclear if ACs are relevant for this computation.

R = We recognized that the previous wording did not clearly convey the relevance of amacrine cells in motion anticipation. To address this, we have added a paragraph describing the study by Johnston & Lagnado (2015), which demonstrated that feedforward inhibition from amacrine cells plays a crucial role in motion anticipation in the zebrafish retina:

Line 1019: *'Although the exact cell type responsible for motion anticipation remains unknown, ACs are strong candidates. Electrophysiological recordings and modeling in goldfish retinas demonstrated that motion anticipation relies on feedforward inhibition from ACs (Johnston & Lagnado, 2015). This inhibition interacts non-linearly with excitatory inputs in GC dendrites, shifting their responses forward in time to encode the actual position of a moving rather than its delayed representation. Inhibitory input is essential for suppressing trailing excitation, ensuring that the stimulus is accurately encoded as it moves across the receptive field.'*

Dear Dr Calbiague,

Re: JP-TR-2025-287958R1 "**The Mysterious Middlemen Making Your Vision Pop: Understanding the Function of Amacrine Cells**" by Victor Calbiague, Deborah Varro, Thomas Buffet, and Olivier Marre

We are pleased to tell you that your paper has been accepted for publication in The Journal of Physiology.

Authors should note that it is too late at this point to offer corrections prior to proofing. Major corrections at proof stage, such as changes to figures, will be referred to the Editors for approval before they can be incorporated. Only minor changes, such as to style and consistency, should be made at proof stage. Changes that need to be made after proof stage will usually require a formal correction notice.

Yours sincerely,

Laura Bennet
Senior Editor
The Journal of Physiology

P.S. - You can help your research get the attention it deserves! Check out Wiley's free Promotion Guide for best-practice recommendations for promoting your work at www.wileyauthors.com/eeo/guide. You can learn more about Wiley Editing Services which offers professional video, design, and writing services to create shareable video abstracts, infographics, conference posters, lay summaries, and research news stories for your research at www.wileyauthors.com/eeo/promotion.

IMPORTANT NOTICE ABOUT OPEN ACCESS: To assist authors whose funding agencies mandate public access to published research findings sooner than 12 months after publication, The Journal of Physiology allows authors to pay an Open Access (OA) fee to have their papers made freely available immediately on publication.

You can check if your funder or institution has a Wiley Open Access Account here: <https://authorservices.wiley.com/author-resources/Journal-Authors/licensing-and-open-access/open-access/author-compliance-tool.html>.

EDITOR COMMENTS

Reviewing Editor:

Dear authors,

Both reviewers are fully satisfied with the changes you made and find the manuscript improved.

Best wishes,

Your Reviewing Editor

REFeree COMMENTS

Referee #1:

The authors have responded nicely to the comments. I have no further concerns.

Referee #2:

The authors address all my comments. Congratulations on doing a great job summarizing a broad topic.